# TRAINING-FREE UNCERTAINTY GUIDANCE FOR COMPLEX VISUAL TASKS WITH MLLMS

## ABSTRACT

Multimodal Large Language Models (MLLMs) often struggle with fine-grained perception, such as identifying small objects in high-resolution images or finding key moments in long videos. Existing works typically rely on complicated, task-specific fine-tuning, which limits their generalizability and increases model complexity. In this work, we propose an effective, training-free framework that uses an MLLM's intrinsic uncertainty as a proactive guidance signal. Our core insight is that a model's output entropy decreases when presented with relevant visual information. We introduce a unified mechanism that scores candidate visual inputs by response uncertainty, enabling the model to autonomously focus on the most salient data. We apply this simple principle to three complex visual tasks: Visual Search, Long Video Understanding, and Temporal Grounding, allowing off-the-shelf MLLMs to achieve performance competitive with specialized, fine-tuned methods. Our work validates that harnessing intrinsic uncertainty is a powerful, general strategy for enhancing fine-grained multimodal performance.

## 1 INTRODUCTION

Multimodal Large Language Models (MLLMs) have achieved remarkable success in general visual understanding, yet they often fall short on tasks that demand fine-grained or localized perception (Wu & Xie, 2024; Zou et al., 2024; Wu et al., 2025a). A critical bottleneck lies in their ability to identify sparse but essential information within vast and noisy visual contexts. This challenge is particularly acute in tasks such as **Visual Search**, which requires locating small objects in high-resolution images; **Long Video Understanding**, which depends on finding key moments in lengthy footage; and **Temporal Grounding**, which involves pinpointing the exact duration of an event. In all these cases, the enormous volume of visual data makes it impractical for models to process everything with equal attention, often leading to overlooked details and incorrect conclusions.

To address this, existing approaches often resort to complex and task-specific solutions. Many rely on computationally intensive methods, such as fine-tuning a reinforcement learning agent to select relevant regions, or integrating external, specialized models for object detection (Zhang et al., 2025b; Zheng et al., 2025; Li et al., 2025). Although sometimes effective, their high training costs and specificity limit generalizability, making them difficult to adapt across diverse tasks requiring nuanced visual reasoning. The core challenge remains: finding a simple, unified, and training-free strategy to guide MLLMs' focus toward the most relevant visual information.

In this work, we propose a simpler alternative, leveraging the MLLM's intrinsic uncertainty. While prior work has shown that a model's output uncertainty, often measured by entropy, correlates with hallucinations and can be used for post-hoc error detection (Xiao & Wang, 2021; Kadavath et al., 2022; Wang et al., 2024), its potential as a proactive guidance signal has been largely unexplored. We posit that this intrinsic uncertainty is a powerful, real-time signal for identifying the most informative visual input. Our key insight is that when an MLLM is presented with the precise visual evidence relevant to a query, its predictive confidence sharpens, and its output uncertainty naturally decreases.

Building on this insight, we introduce the **Uncertainty-Guided (UG) framework**, a novel approach that reformulate these distinct localization challenges as a single, unified problem: searching for the state of minimum uncertainty. Our framework implements this by systematically evaluating candidate visual inputs (e.g., image crops, video frames, or temporal windows) and scoring each based on the MLLM's response uncertainty. This uncertainty can be quantified directly via the full

output distribution's *entropy* or, for binary decisions, through a more direct *confidence score* based on "yes/no" probabilities. This simple mechanism allows the model to autonomously identify and focus on the most salient information. Regardless of whether the task is to pick the most relevant crop for visual search, sample the top-k informative frames for video QA, or identify the segment with the highest confidence for temporal grounding, the underlying principle remains unchanged.

This unified, training-free, and model-agnostic framework allows any off-the-shelf MLLM to solve complex localization tasks without architectural modifications or fine-tuning. We apply it to various open-sourced models, including the LLaVA (Li et al., 2024), Qwen-VL (Bai et al., 2025), and InternVL (Chen et al., 2024c) families. Our contributions are summarized as follows: (1) We empirically establish a strong inverse correlation between MLLM output uncertainty and performance, validating uncertainty as a reliable, real-time signal for visual relevance. (2) We propose a simple and unified training-free framework that leverages intrinsic uncertainty to solve diverse, fine-grained visual localization tasks without task-specific designs. (3) Comprehensive experiments show that our framework enables standard MLLMs to achieve competitive results against state-of-the-art specialized methods on Visual Search, Long Video Understanding, and Temporal Grounding.

## 2 RELATED WORK

**Multimodal Large Language Models (MLLMs).** The field of MLLMs has rapidly evolved into highly capable systems integrating perception and reasoning. Foundational open-source models like LLaVA (Liu et al., 2024; Li et al., 2024), Qwen-VL (Bai et al., 2023; 2025), and InternVL (Chen et al., 2024d;c) effectively align visual features with LLMs. This paradigm has been extended from static images to video with models such as LLaVA-Video (Zhang et al., 2024b) and Intern-Video (Wang et al., 2022). While proficient at general visual question answering, these models often struggle with tasks requiring a deep, localized understanding of specific visual details.

**Visual Limitations in MLLMs.** Despite advances, MLLMs face inherent fine-grained perception limitations. First, they struggle with small objects, as fixed, low-resolution visual encoders can lose critical details during feature extraction (Wu & Xie, 2024; Zhang et al., 2025a). Second, long-video understanding is challenging due to constrained LLM context lengths, which prevents processing all frames and forces reliance on sub-optimal sampling strategies (Zou et al., 2024). Finally, most MLLMs lack precise temporal grounding, as they are not natively trained with explicit time tokens or time-aware datasets, thus requiring specialized fine-tuning to acquire this skill (Huang et al., 2024; Chen et al., 2024a). Our work addresses these three challenges within a single, unified framework.

**Uncertainty in MLLMs.** Uncertainty quantification is a cornerstone of building reliable machine learning systems. In both LLMs and MLLMs, entropy serves as a key indicator of model uncertainty, where higher values often correlate with hallucinations and incorrect predictions (Kadavath et al., 2022; Chen et al., 2024b; Leng et al., 2024). While previous work has primarily used this for post-hoc hallucination detection, we propose a novel, training-free framework that instead uses entropy as a proactive signal to actively guide the model's decision-making process on complex visual tasks.

## 3 THE UNCERTAINTY-GUIDED (UG) FRAMEWORK

In this section, we detail our proposed Uncertainty-Guided (UG) framework, a training-free approach that enhances MLLM performance on fine-grained perception tasks. We begin by establishing the core principle that supports our entire method: actively minimizing a model's intrinsic uncertainty is an effective strategy for localizing salient visual input. After empirically validating this principle, we formalize the uncertainty metrics used for scoring visual inputs and then detail the application of our simple "score-then-answer" mechanism to three distinct localization challenges.

### 3.1 PRINCIPLE: MINIMIZING UNCERTAINTY TO LOCALIZE KEY VISUAL INFORMATION

Our framework is built on the hypothesis that an MLLM's intrinsic uncertainty can be actively minimized to *localize the key visual information* required for a correct answer. While prior work uses output entropy for post-hoc error detection, we propose it as a proactive signal for localization. To validate this, we conducted a motivating experiment on $V^*$ Bench (Wu & Xie, 2024), which features fine-grained questions. We divided the high-resolution image into several crops, each with

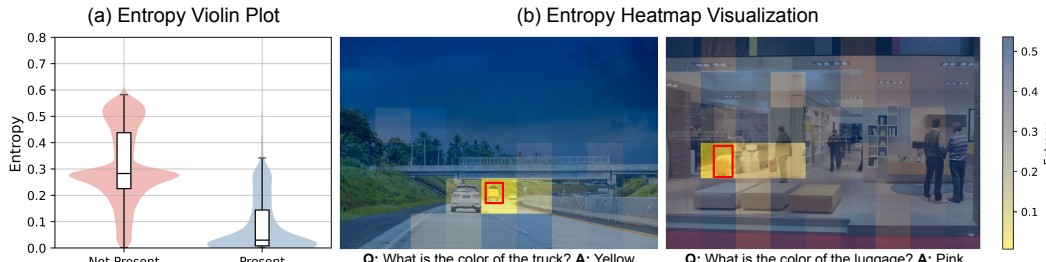

Figure 1: (a) Entropy violin plot comparing visual crops that include a target object (`Present`, e.g., the truck) versus those that do not (`Not Present`, e.g., background). (b) Entropy heatmap showing how entropy scores vary across different visual crops. Red box denotes the target object.

a size of $1/6 \min(\text{width}, \text{height})$, and fed each to LLaVA-OneVision-7B (Li et al., 2024) with the original query, hypothesizing that crops containing the salient information would yield *confident (low-entropy)* predictions, while irrelevant crops would produce *uncertain (high-entropy)* ones.

The results, illustrated in Figure 1, strongly support this hypothesis. Figure 1(a) shows a clear, statistically significant separation in the entropy distributions: `Present` crops with the target object (e.g., the truck) exhibit a distribution shifted toward lower entropy compared to `Not Present` crops (e.g., background). This local effect is visualized in Figure 1(b), where the crops over the query's target (e.g., truck and luggage) yield the lowest entropy scores (yellow). This strong inverse correlation confirms that an MLLM's output uncertainty can serve as a reliable proxy for visual-information relevance. This foundational observation allows us to reframe complex fine-grained problems as a search for the visual input that *minimizes the model's uncertainty*. Further analysis on this correlation and theoretical discussion can be found in Appendix B and C.

## 3.2 UNCERTAINTY AS A SCORING FUNCTION

At the heart of our framework is the ability to quantify the model's uncertainty for a given visual input $\boldsymbol{v}$ and textual query $\boldsymbol{q}$. We employ two distinct metrics derived from the output probability distribution, $\boldsymbol{p}_i$, generated by the model's vocabulary head at each step $i$. For general-purpose relevance scoring, we use **Token Entropy**. We calculate the standard Shannon entropy (Shannon, 1948) for each generated token's distribution and average this value across the entire generated sequence of length $T$. This provides a holistic measure of the model's overall confidence:

$$\mathcal{H}(\boldsymbol{v}, \boldsymbol{q}) = -\frac{1}{T} \sum_{i=1}^{T} \sum_{j=1}^{N} p_{i,j} \log(p_{i,j}) \tag{1}$$

where $N$ is the vocabulary size and $p_{i,j}$ is the probability of the $j$th vocabulary token at step $i$. A lower entropy signifies lower uncertainty and thus higher relevance of the visual input to the query.

For tasks that require a binary decision (e.g., "is an event present?"), token entropy is less suitable, as high confidence in either "yes" or "no" answer would produce low entropy. For these cases, we use a more targeted metric named **Binary Response Confidence (BRC)** score. It directly measures the model's certainty toward a positive confirmation by taking the difference between the probabilities of the "yes" and "no" tokens in the first generated distribution $p_1$:

$$S_{\text{conf}}(\mathbf{v}, \mathbf{q}) = p_1(\text{"yes"}) - p_1(\text{"no"}) \tag{2}$$

This score provides a directional measure of uncertainty, where a high positive value indicates strong confidence in the event's presence.

## 3.3 APPLICATIONS OF THE UG FRAMEWORK

The UG framework's simple "score-then-answer" mechanism is general enough to be applied to various localization tasks including Visual Search, Long Video Understanding, and Temporal Grounding as shown in Figure 2.

**UG-Search: Visual Search in High-Resolution Images.** To find small objects in large images, UG-Search systematically divides the image into a set of candidate crops using a sliding window.

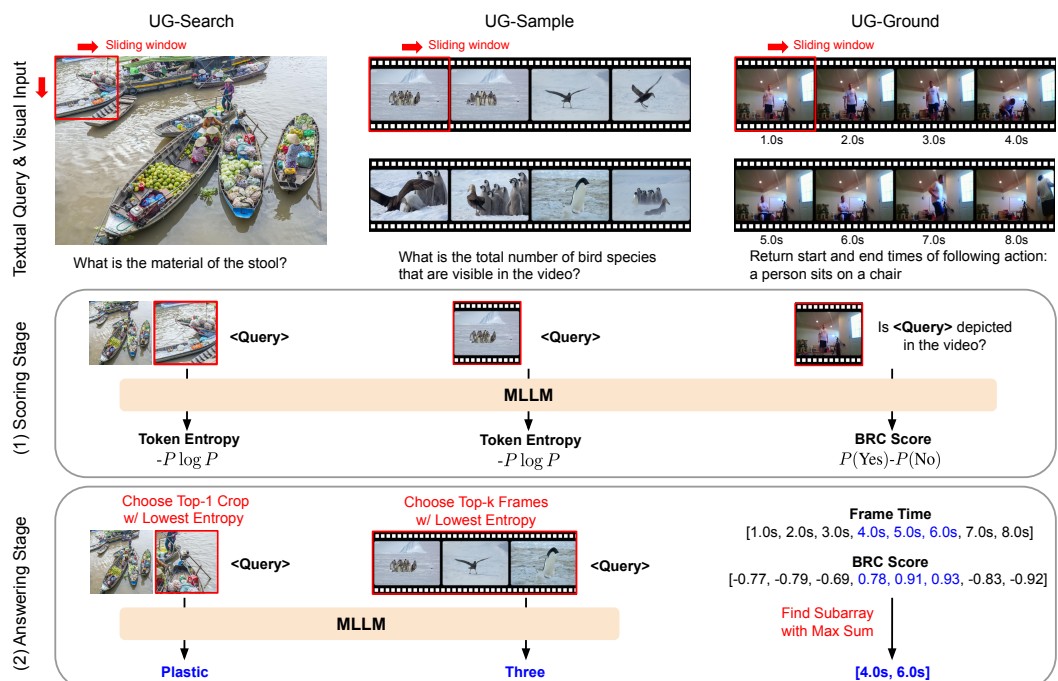

Figure 2: **An overview of our Uncertainty-Guided (UG) Framework.** The framework follows a two-stage process: (1) **Scoring Stage:** Candidate visual inputs (image crops or video frames) are scored using the MLLM's intrinsic uncertainty, measured by either Token Entropy or Binary Response Confidence (BRC) score. (2) **Answering Stage:** The input with the lowest uncertainty are used for a final inference to generate the definitive answer.

Each crop is scored by feeding it, along with the original image, to the MLLM and calculating its average **Token Entropy** (Eq. 1). This initial inferences are solely for scoring. The single crop that yields the lowest entropy is identified as the most informative region. The final high-quality answer is then based on this selected crop only.

**UG-Sample: Frame Sampling for Long Videos.** We extend the same principle to long video understanding. To find the most relevant moments, UG-Sample treats each frame (or short window) as a candidate visual input. Each candidate is scored using its **Token Entropy**. Then, the top-$k$ frames with the lowest entropy are selected, combined into a single context, and used for final inference to answer the query.

**UG-Ground: Temporal Grounding of Events.** To find the precise start and end times of an action, UG-Ground reframes the task as a search for the most confident contiguous temporal segment. It uses a sliding window to process the video, scoring each with the **BRC score** (Eq. 2) by querying whether the target event is depicted. This converts the video into a sequence of confidence scores. Temporal grounding is thus reduced to finding the subarray with the maximum sum which can be solved efficiently in linear time using *Kadane's Algorithm* (Bentley, 1984). The start and end indices of this subarray directly correspond to the predicted timestamps.

## 4 EXPERIMENTS

We conduct a comprehensive set of experiments to validate the effectiveness, generality, and scalability of our Uncertainty-Guided (UG) framework. We evaluate its performance on three distinct and challenging localization tasks: Visual Search (Section 4.1), Video Frame Sampling (Section 4.2), and Video Temporal Grounding (Section 4.3). Several analytical experiments are shown in Section 4.4 and how to improve efficiency is discussed in Section 4.5. To ensure fair and reproducible comparisons, all evaluations are performed using the *LMMs-Eval* (Zhang et al., 2024a). Implementation details are available in Appendix F.

Table 1: **Results of UG-Search on High Resolution Image Benchmarks.** The **bold** number represents the best model, while the underscored number represents the second-best model. The value in subscript ($\Delta$) indicates the performance change from the baseline (green for gains, red for losses).

| Model | $V^*$ Bench | | | HR-Bench 4K | | | HR-Bench 8K | | |
|---|---|---|---|---|---|---|---|---|---|
| | Attribute | Spatial | Overall | FSP | FCP | Overall | FSP | FCP | Overall |
| GPT-4o | 72.2 | 60.5 | 67.5 | 66.8 | 63.3 | 65.0 | 60.8 | **58.5** | 59.6 |
| Gemini 1.5 Flash | - | - | - | 76.8 | 56.8 | 66.8 | 69.2 | 56.7 | 62.8 |
| SEAL-7B | 74.8 | 76.3 | 75.4 | 47.0 | 29.3 | 38.1 | 42.5 | 28.8 | 35.6 |
| DyFo-7B | 80.0 | 82.9 | 81.2 | - | - | - | - | - | - |
| DeepEyes-7B | 91.3 | **88.2** | 90.1 | **91.3** | 59.0 | 75.1 | 86.8 | **58.5** | 72.6 |
| LLaVA-OV-7B | 79.1 | 67.1 | 74.4 | 74.3 | 55.5 | 64.9 | 65.0 | 51.8 | 58.4 |
| w/ UG-Search$_\Delta$ | 94.8$_{15.7}$ | 75.0$_{7.9}$ | 86.9$_{12.5}$ | 84.8$_{10.5}$ | 55.5$_{0.0}$ | 70.1$_{5.2}$ | 86.3$_{21.3}$ | 51.5$_{0.3}$ | 68.9$_{10.5}$ |
| Qwen2.5-VL-7B | 78.3 | 73.7 | 76.4 | 83.3 | 58.8 | 71.0 | 78.8 | 52.0 | 65.4 |
| w/ UG-Search$_\Delta$ | 91.3$_{13.0}$ | 76.3$_{2.6}$ | 85.3$_{8.9}$ | 90.5$_{7.2}$ | 58.0$_{0.8}$ | 74.3$_{3.3}$ | 87.0$_{8.2}$ | 51.5$_{0.5}$ | 69.3$_{3.9}$ |
| InternVL2.5-8B | 71.3 | 72.4 | 71.7 | 63.3 | 56.3 | 59.8 | 62.0 | 54.0 | 58.0 |
| w/ UG-Search$_\Delta$ | **97.4**$_{26.1}$ | 81.6$_{9.2}$ | **91.1**$_{19.4}$ | 91.0$_{17.7}$ | 60.5$_{4.2}$ | **75.8**$_{16.0}$ | **90.3**$_{28.3}$ | 57.0$_{3.0}$ | **73.6**$_{15.6}$ |

Table 2: **Comparison with State-of-the-Art Visual Search Methods.**

| | $V^*$ | HR4K | HR8K |
|---|---|---|---|
| LLaVA-OV-7B | 74.4 | 64.9 | 58.4 |
| w/ ZoomEye | 85.0 | 68.4 | 66.5 |
| w/ UG-Search | **86.9** | **70.1** | **68.9** |
| Qwen2.5-VL-7B | 64.9 | 60.1 | 53.1 |
| w/ TextCoT | 67.0 | 60.6 | 50.6 |
| w/ Rel-att | 69.6 | 63.3 | 51.9 |
| w/ Thyme | 68.1 | 66.6 | 59.0 |
| w/ UG-Search | **81.7** | **74.9** | **66.3** |

Table 3: **Comparison with State-of-the-Art Video Frame Sampling Methods.**

| | V-MME | MLVU | LVB |
|---|---|---|---|
| LLaVA-OV-7B | 53.9 | 58.6 | 54.3 |
| w/ KFC | 55.4 | **65.0** | 55.6 |
| w/ BOLT | 56.1 | 63.4 | 55.6 |
| w/ AKS | 56.1 | 64.8 | 56.8 |
| w/ FRAG | 56.3 | 64.9 | 57.3 |
| w/ UG-Sample | 58.6 | 62.0 | **59.5** |
| w/ UG-Sample+AKS | 59.2 | 65.0 | 59.4 |

## 4.1 UG-SEARCH: UNCERTAINTY-GUIDED VISUAL SEARCH

**Setup.** We evaluate UG-Search on two demanding high-resolution image benchmarks: $V^*$ Bench (Wu & Xie, 2024) and HR-Bench (Wang et al., 2025a), which contain images with resolutions from 2k to 8k. For each benchmark, we report performance on distinct splits. For $V^*$ Bench, we evaluate on Attribute and Spatial reasoning. For HR-Bench, we report results on Fine-grained Single-instance Perception (FSP) and Fine-grained Cross-instance Perception (FCP).

Our method processes the image with a square sliding window; the crop size is set to one-sixth of the image's smaller dimension, with a stride of half the crop size. We apply our training-free UG-Search to several open-source MLLMs, including LLaVA-OneVision (Li et al., 2024), Qwen2.5-VL (Bai et al., 2025), and InternVL-2.5 (Chen et al., 2024c). We compare its performance against a comprehensive set of baselines, which we group into three categories: (1) *Proprietary Models*, such as GPT-4o (Hurst et al., 2024) and Gemini 1.5 Flash (Team et al., 2024); (2) *Training-Free Methods*, like ZoomEye (Shen et al., 2025), DyFo (Li et al., 2025), Rel-att (Zhang et al., 2025a), and TextCoT (Luan et al., 2024); and (3) *Specialized Fine-Tuned Methods*, which include Deep-Eyes (Zheng et al., 2025), Thyme (Zhang et al., 2025b), and SEAL (Wu & Xie, 2024).

**Results and Analysis.** As shown in Table 1, applying UG-Search yields substantial and consistent performance gains across all base models. The improvements are particularly striking on tasks requiring single-object perception (the Attribute and FSP splits), where accuracy increases by as much as +26.1% for InternVL2.5-8B on $V^*$ Bench. This result powerfully demonstrates our method's ability to autonomously guide the MLLM's focus to the correct, fine-grained region of interest. Tasks involving spatial relationships between multiple objects (Spatial and FCP splits) prove more challenging, as they demand both precise object localization and complex reasoning. In Appendix I, we qualitatively analyze failure cases for relational reasoning to better understand these results.

In Table 2, we directly compare UG-Search against specialized methods built on the same pretrained MLLMs. Our approach surpasses ZoomEye, another training-free method, across all benchmarks. Methods like TextCoT and Rel-att, which use the MLLM to generate bounding boxes or rely on attention maps, show limited gains on higher-resolution benchmarks like HR-Bench. Thyme, which employs supervised fine-tuning (SFT) and reinforcement learning (RL) to enable the MLLM to gen-

Table 4: **Results of UG-Sample on Long Video Understanding Benchmarks.** The **bold** number represents the best model, while the underscored number represents the second-best model. The value in subscript ($\Delta$) indicates the improvement over the baseline.

| Model | Video-MME (w/o sub.) | | | | $\mathbf{MLVU}_{dev}$ | $\mathbf{LVB}_{val}$ |
| --- | --- | --- | --- | --- | --- | --- |
| | Overall | Short | Medium | Long | | |
| *Video Length* | *17min* | *1.3min* | *9min* | *41min* | *12min* | *8min* |
| Frame-Voyager-7B | 57.5 | 67.3 | 56.3 | 48.9 | 65.6 | - |
| LLaVA-OV-7B | 53.9 | 64.9 | 52.1 | 44.7 | 58.6 | 54.3 |
| w/ **UG-Sample**$_\Delta$ | $58.6_{4.7}$ | $69.7_{4.8}$ | $58.4_{6.3}$ | $47.7_{3.0}$ | $62.0_{3.4}$ | $\mathbf{59.5}_{5.2}$ |
| LLaVA-Video-7B | 55.9 | 67.7 | 53.6 | 46.4 | 56.4 | 55.7 |
| w/ **UG-Sample**$_\Delta$ | $\underline{59.7}_{3.8}$ | $\mathbf{70.6}_{2.9}$ | $\mathbf{60.1}_{6.5}$ | $48.3_{1.9}$ | $61.9_{5.5}$ | $58.1_{2.4}$ |
| InternVL2.5-8B | 57.8 | 68.1 | 56.4 | 48.8 | 61.6 | 52.8 |
| w/ **UG-Sample**$_\Delta$ | $\mathbf{60.6}_{2.8}$ | $\underline{70.3}_{2.2}$ | $\underline{59.9}_{3.5}$ | $\mathbf{51.7}_{2.9}$ | $\mathbf{67.6}_{6.0}$ | $\underline{59.5}_{6.7}$ |
| InternVideo2.5-8B | 54.2 | 62.9 | 52.7 | 47.0 | 59.4 | 51.2 |
| w/ **UG-Sample**$_\Delta$ | $57.4_{3.2}$ | $65.8_{2.9}$ | $56.0_{3.3}$ | $\underline{50.3}_{3.3}$ | $63.2_{3.8}$ | $57.7_{6.5}$ |

erate cropping code, requires extensive training. In contrast, our training-free UG-Search achieves superior performance, highlighting the effectiveness of our uncertainty-based approach. In Appendix H.1, we found UG-Search also improves the performance of standard VQA tasks.

## 4.2 UG-SAMPLE: UNCERTAINTY-GUIDED VIDEO FRAME SAMPLING

**Setup.** We next evaluate our framework's ability to identify key query-relevant moments within long and complex videos. We use three standard long-video benchmarks: Video-MME (Fu et al., 2025), MLVU (Zhou et al., 2025), and LongVideoBench (Wu et al., 2024), which feature videos with durations ranging from several minutes to over an hour. To strictly isolate the contribution of the video sampling strategy, all experiments on Video-MME are conducted without subtitles.

Referring to established protocols (Liu et al., 2025; Huang et al., 2025), we select the top 8 frames from a pool of 256 uniformly sampled candidate frames, using a window size of one frame for entropy calculation. For baseline models, 8 frames are uniformly sampled over the entire video. We apply UG-Sample to four video-capable MLLMs, including both versatile models (LLaVA-OneVision (Li et al., 2024), InternVL-2.5 (Chen et al., 2024c)) and those specifically optimized for video (LLaVA-Video (Zhang et al., 2024b), InternVideo-2.5 (Wang et al., 2025b)). We benchmark against a variety of state-of-the-art sampling methods that use different underlying principles: (1) *VLM-based methods* that rely on external visual-text similarity scores (BOLT (Liu et al., 2025), AKS (Tang et al., 2025)); (2) a *graph-based optimization* method (KFC (Fang et al., 2025)); and (3) *MLLM-based methods* that either use binary questions to score frames (FRAG (Huang et al., 2025)) or train a reward function based on MLLM embeddings (Frame-Voyager (Yu et al., 2024)).

**Results and Analysis.** As shown in Table 4, applying UG-Sample provides consistent performance improvements over the standard uniform sampling baseline across all four MLLMs. Notably, our method enhances not only general-purpose models but also those already optimized for video tasks, demonstrating the universal benefit of uncertainty-guided frame selection. Notably, LLaVA-OneVision's accuracy on Video-MME improves from 53.9% to 58.6%. The comparison against state-of-the-art methods in Table 3 further highlights the strength of our approach. UG-Sample on its own is highly competitive, outperforming several specialized baselines. We also created a hybrid method, UG-Sample+AKS, by replacing the CLIP-based scoring function in AKS with our entropy score. This hybrid model achieves the best performance on two of the three benchmarks. This result strongly suggests that an MLLM's intrinsic uncertainty is a more semantically rich signal for frame relevance than external CLIP similarity scores or the repeated binary prompting used by FRAG. In Appendix H.2, we further demonstrate that UG-Sample is also effective for relatively short videos.

## 4.3 UG-GROUND: UNCERTAINTY-GUIDED TEMPORAL GROUNDING

**Setup.** Finally, we evaluate our framework on video temporal grounding, challenging task of identifying the precise start and end times of a specific event within a video. We use two standard benchmarks, Charades-STA (Gao et al., 2017) and ActivityNet Captions (Krishna et al., 2017). Fol-

Table 5: **Results of UG-Ground on Video Temporal Grounding Benchmarks.** The **bold** number represents the best model, while the underscored number represents the second-best model. The value in subscript ($\Delta$) indicates the improvement over the baseline.

| Model | Charades-STA | | | | ActivityNet Captions | | | |
|---|---|---|---|---|---|---|---|---|
| | R@0.3 | R@0.5 | R@0.7 | mIoU | R@0.3 | R@0.5 | R@0.7 | mIoU |
| TimeChat-7B | 40.6 | 23.8 | 9.7 | 26.2 | 25.0 | 13.2 | 6.1 | 18.5 |
| VTimeLLM-7B | 51.0 | 27.5 | 11.4 | 31.2 | 44.0 | 27.8 | 14.3 | 30.4 |
| VTimeLLM-13B | 55.3 | 34.3 | 14.7 | 34.6 | 44.8 | 29.5 | 14.2 | 31.4 |
| TimeMarker-8B | 73.5 | 51.9 | 26.9 | 48.4 | - | - | - | - |
| VTG-GPT | 59.5 | 43.7 | 25.9 | 39.8 | 47.1 | 28.3 | 12.8 | 30.5 |
| TFVTG | 67.0 | 50.0 | 24.3 | 44.5 | 49.3 | 27.0 | 13.4 | 34.1 |
| TAG | 67.8 | 48.6 | 26.7 | 45.7 | 51.9 | 28.9 | 15.1 | 36.6 |
| LLaVA-OV-7B | 14.4 | 7.2 | 3.5 | 10.4 | 14.8 | 7.2 | 3.0 | 11.9 |
| w/ **UG-Ground**$_\Delta$ | $69.0_{54.6}$ | $45.7_{38.5}$ | $25.9_{22.4}$ | $46.8_{36.4}$ | $53.4_{38.6}$ | $29.2_{22.0}$ | $15.5_{12.5}$ | $37.2_{25.3}$ |
| LLaVA-Video-7B | 16.8 | 8.3 | 3.0 | 11.4 | 20.9 | 9.9 | 3.9 | 15.4 |
| w/ **UG-Ground**$_\Delta$ | $67.5_{50.7}$ | $47.2_{38.9}$ | $29.2_{26.4}$ | $47.2_{35.8}$ | $54.9_{34.0}$ | $30.8_{20.9}$ | $16.5_{12.6}$ | $38.3_{22.9}$ |
| InternVL2.5-8B | 11.7 | 4.1 | 1.9 | 9.5 | 6.0 | 2.4 | 1.0 | 5.7 |
| w/ **UG-Ground**$_\Delta$ | $67.6_{55.9}$ | $49.3_{45.2}$ | $30.4_{28.5}$ | $47.4_{37.9}$ | $52.7_{46.7}$ | $30.0_{27.6}$ | $16.2_{15.2}$ | $36.9_{31.2}$ |
| InternVideo2.5-8B | 50.2 | 32.0 | 14.4 | 32.3 | 18.9 | 9.5 | 4.1 | 14.0 |
| w/ **UG-Ground**$_\Delta$ | $\mathbf{74.7}_{24.5}$ | $\mathbf{52.5}_{20.5}$ | $\mathbf{32.6}_{18.2}$ | $\mathbf{51.0}_{18.7}$ | $54.9_{36.0}$ | $\mathbf{30.8}_{21.3}$ | $16.4_{12.3}$ | $\mathbf{38.5}_{24.5}$ |

lowing previous evaluation schemes (Lee et al., 2025; Zheng et al., 2024), we report performance using standard recall at various IoU thresholds (R@k) and mean Intersection over Union (mIoU).

For implementation, we calculate the BRC score using a sliding window of 15 frames with a stride of 1 frame. Input videos are sampled at 3 FPS for Charades-STA and 1 FPS for ActivityNet Captions to manage computational cost. We apply our UG-Ground method to the same four video-capable MLLMs and compare the results against two distinct categories of highly specialized, state-of-the-art methods: (1) *Instruction-Tuned Models*, such as VTimeLLM (Huang et al., 2024), TimeChat (Ren et al., 2024), and TimeMarker (Chen et al., 2024a), which are explicitly fine-tuned with temporal reasoning data or time-aware tokens; and (2) *Training-Free Methods*, like VTG-GPT (Xu et al., 2024), TFVTG (Zheng et al., 2024), and TAG (Lee et al., 2025), which rely on an external, pretrained VLM to generate image caption or compute frame-query similarity scores.

**Results and Analysis.** As shown in Table 5, standard MLLMs are largely incapable of performing temporal grounding out of the box, reflecting that they are not trained for such precise localization. However, applying our UG-Ground framework transforms their capabilities, yielding massive performance improvements across all models and benchmarks. For instance, LLaVA-OneVision's mIoU on Charades-STA increases dramatically from 10.4 to 46.8 (+36.4), demonstrating the impressive impact of our uncertainty-guided approach.

Remarkably, this simple, training-free method elevates general-purpose MLLMs to a level where they consistently outperform models specifically instruction-tuned for temporal tasks (VTimeLLM, TimeChat, and TimeMarker). Furthermore, UG-Ground is more effective than other training-free approaches like TAG. This comparison is particularly insightful, as it shows that leveraging the primary MLLM's own internal confidence signal (via our BRC score) is a more powerful and direct strategy than relying on similarity scores computed by an external model. By applying UG-Ground to a video-specialized model like InternVideo2.5-8B, we achieve state-of-the-art results, solidifying the power and versatility of our unified framework.

## 4.4 UG FRAMEWORK ANALYSIS

In this section, we conduct a series of analytical experiments to better understand the properties of our Uncertainty-Guided (UG) framework. We investigate the effect of input granularity, analyze the framework's scalability with model size, and show qualitative results on three distinct tasks.

**Ablation Study on Input Granularity and Inference Time** We investigate how visual input granularity (crop size for images, window size for videos) affects both performance (accuracy) and efficiency (inference time). We measure average inference time in seconds per example on NVIDIA

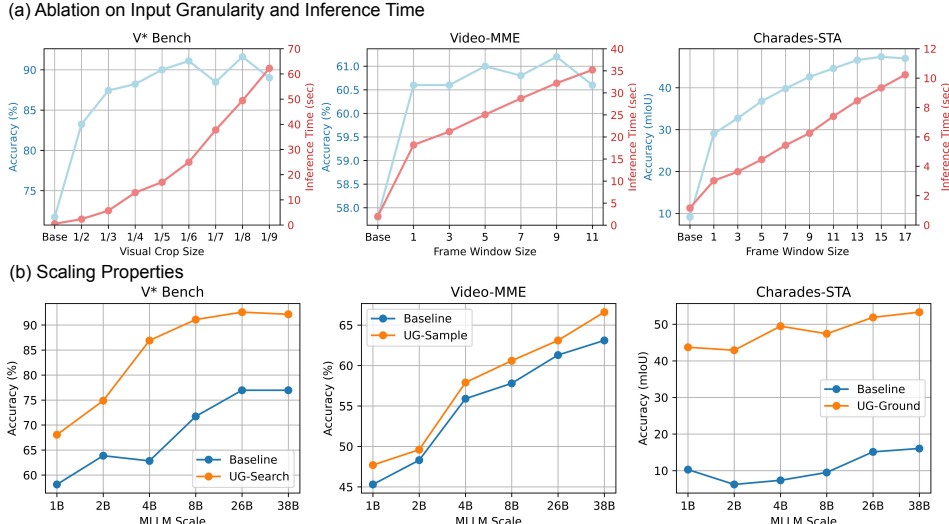

Figure 3: (a) Performance as a function of visual crop size (for UG-Search) or frame window size (for UG-Sample and UG-Ground) with corresponding inference time. `Base` represents baseline. (b) Performance of the baseline and our UG-enhanced models across the InternVL-2.5 family.

A100 80G GPUs. As shown in Figure 3(a), using InternVL2.5-8B, we observe a consistent and critical trade-off between task performance and computational cost.

For UG-Search on $V^*$ Bench, accuracy (light blue line) peaks at a fine granularity (crop size $1/8$), though performance gains begin to saturate after $1/6$. Concurrently, inference time (red line) rises as granularity becomes finer, increasing from $\sim 2$ seconds at $1/2$ to over 60 seconds at $1/9$. This clearly demonstrates that pursuing marginal accuracy gains at very fine granularities incurs a disproportionate computational cost. For the video tasks, the optimal granularity is task-dependent. UG-Ground on Charades-STA (action understanding) benefits from richer temporal context, with accuracy peaking at a longer 15-frame window. In contrast, for UG-Sample on Video-MME (video QA), a single-frame window provides near-peak performance, suggesting that identifying discrete, relevant moments is more critical than analyzing long segments. In both video tasks, inference time shows a steady, linear increase with window size.

Overall, UG-method performance consistently outperforms the baseline at all granularities, demonstrating robustness. In practice, this provides a clear control for users: a coarser input (e.g., $1/2$ crop, 1-frame window) can be chosen for high-speed inference, while a finer input (e.g., $1/8$ crop, 15-frame window) can be used to achieve maximum performance, allowing a precise balance between accuracy and computational cost.

**Analysis of Scaling Properties** In Figure 3(b), we examine the scalability of our UG framework by applying it to the InternVL-2.5 family of models, ranging from 1B to 38B parameters. The results show that the performance of both the baseline models and our UG-enhanced models improves consistently as model size increases. Crucially, the performance gains achieved by our UG methods are robust across all model scales, with the largest models achieving the highest absolute accuracy on each benchmark. This consistent improvement suggests that as MLLMs become larger and, presumably, better calibrated, their intrinsic uncertainty becomes an even more reliable signal that our framework can effectively harness.

**Qualitative Results** Figure 4 provides qualitative examples that demonstrate how our UG framework pinpoints relevant information within noisy visual contexts, leading to correct predictions where baseline models fail. The examples are drawn from $V^*$ Bench, Video-MME, and ActivityNet Captions. We provide extended qualitative results in Appendix I.

In the examples for (a), UG-Search overcomes the baseline's tendency to be distracted by cluttered scenes. In the left image, it isolates the woman with the backpack to correctly identify her pose as "Squatting", while the baseline incorrectly guesses a common pose ("Walking"). Similarly, on the right image, it successfully localizes "small red car" and "baby carriage" to accurately resolve

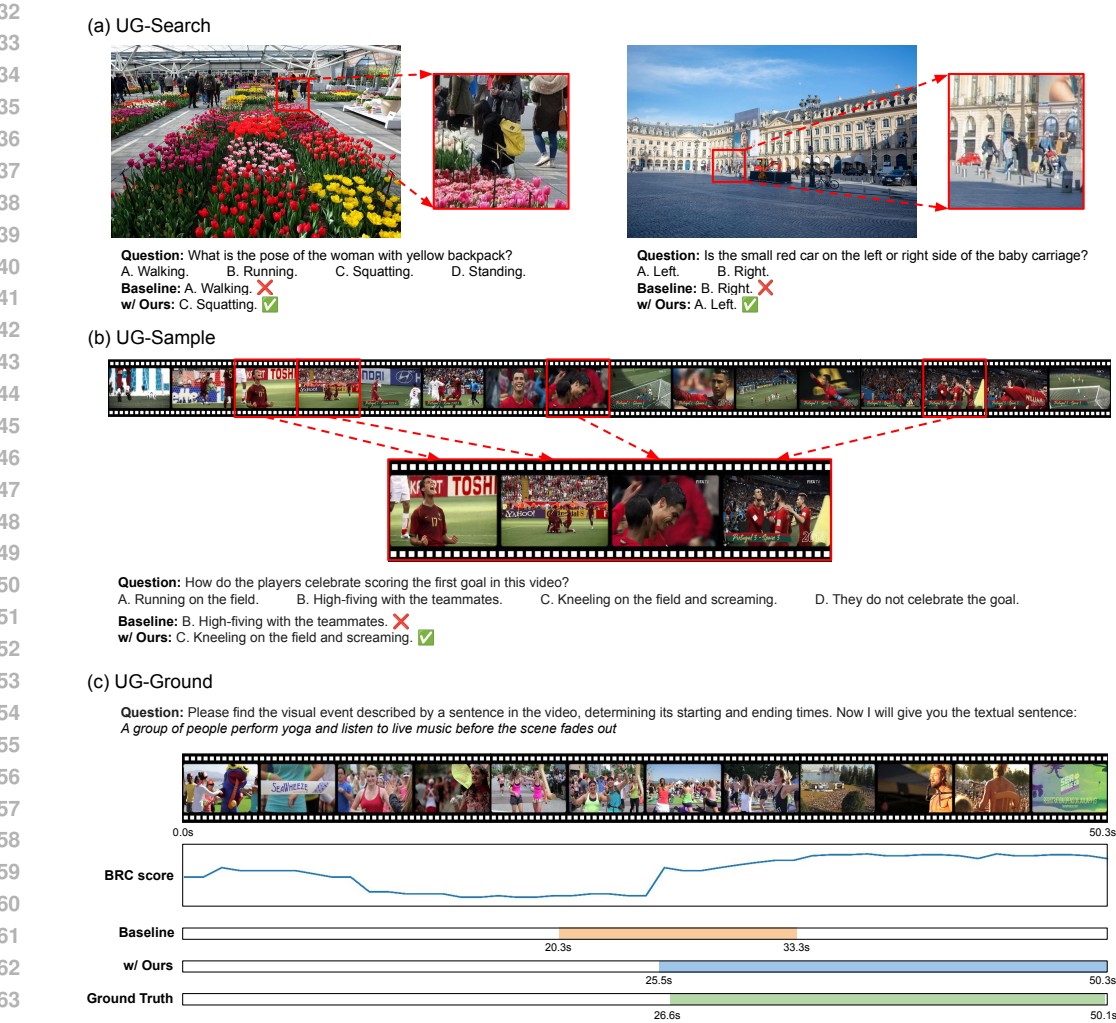

Figure 4: **Qualitative Results.** (a) UG-Search localizes small target objects by selecting the lowest-entropy crop (red box). (b) UG-Sample identifies key semantic frames (red box) with the lowest entropy from a long video. (c) UG-Ground pinpoints the correct event timeline by finding the peak in its BRC score sequence.

their spatial relationship. The video example in (b) shows UG-Sample identifying the precise frames of a goal celebration. The baseline, likely processing irrelevant moments from a uniform sample, fails to understand the event correctly. Our method, however, filters the timeline to provide the necessary context relevant to the query, leading to the correct answer ("C. Kneeling on the field and screaming"). Finally, in (c), the baseline fails to identify the correct time segment for the "yoga and live music" event. Our UG-Ground method first transforms the video into a sequence of BRC scores, which exhibits a clear plateau of high confidence during the correct interval. This allows our method to accurately ground the event's duration, aligning almost perfectly with the ground truth.

## 4.5 IMPROVING THE EFFICIENCY OF UG FRAMEWORK

The primary limitation of our framework is the increased inference time from its multiple scoring inferences. In this section, we explore two practical strategies to mitigate this overhead: (1) decoupling the scorer and generator models and (2) using external vision models as pre-filters.

**Decoupling the Scorer and Generator** To improve efficiency, we investigate decoupling the MLLM used for the uncertainty scoring stage from the one used for final answer generation. As shown in Table 6, this strategy reveals a clear performance-efficiency trade-off. Using a smaller scorer, such as the 1B model, dramatically reduces inference time (e.g., from 24.8s to 10.8s on $V^*$

Bench) compared to using the symmetric 8B scorer. However, this speed gain comes at a significant cost to accuracy (dropping from 91.1% to 75.4%). Conversely, using a larger, more capable scorer consistently improves final performance. Guiding the 8B answering model with a 26B scoring model achieves the highest accuracy on both $V^*$ Bench (92.0%) and Video-MME (61.8%). This decoupling opens promising avenues for designing efficient, cascaded systems, or for future exploration of knowledge distillation to create compact yet powerful scorer models.

Table 6: **Performance with Decoupled Scorer and Generator Models.** Using a larger MLLM for scoring improves accuracy, while a smaller scorer reduces total inference time. All inference times are reported in seconds per example.

| Answering MLLM | Scoring MLLM | UG-Search | | UG-Sample | |
|---|---|---|---|---|---|
| | | $V^*$ Bench | Infer. Time | Video-MME | Infer. Time |
| InternVL2.5-8B | None | 71.7 | 0.6s | 57.8 | 2.0s |
| InternVL2.5-8B | InternVL2.5-1B | 75.4 | **10.8s** | 58.5 | **11.5s** |
| InternVL2.5-8B | InternVL2.5-2B | 79.1 | 15.3s | 59.0 | 13.0s |
| InternVL2.5-8B | InternVL2.5-4B | 89.0 | 19.2s | 60.1 | 16.6s |
| InternVL2.5-8B | InternVL2.5-8B | 91.1 | 24.8s | 60.6 | 18.4s |
| InternVL2.5-8B | InternVL2.5-26B | **92.0** | 40.5s | **61.8** | 27.7s |

**UG Framework with External Pre-filters** Another strategy to reduce runtime is to use lightweight, off-the-shelf vision models as efficient *pre-filters*. These models select a smaller, high-potential subset of candidates, reducing the number of inputs the UG method should score.

As shown in Table 7, we test this for UG-Search using region proposals from YOLOv8 (Varghese & Sambath, 2024) and segmentations from SAM2 (Ravi et al., 2025). The results show a clear trade-off, where our full UG-Search remains the most robust option, achieving the highest accuracy (81.2%) at a moderate cost (14.6s). The pre-filters struggle to find a good balance: the YOLOv8 Pre-filter is faster (9.0s) but its accuracy (74.9%) is only marginally better than the baseline (74.4%). Conversely, the SAM2 Pre-filter improves accuracy (78.2%) but is impractically slow, incurring the highest computational cost (26.5s). This suggests that for complex scenes, generic pre-filters may be an inefficient trade-off compared to our exhaustive search.

In contrast, for UG-Sample in video, pre-filtering proves highly effective. As shown in Table 8, we test a redundancy-based filter (DINOv2 (Oquab et al., 2023)) and a relevance-based filter (SigLIP (Zhai et al., 2023)). The relevance-based SigLIP Pre-filter provides an excellent balance, achieving accuracy (57.3%) comparable to the full UG-Sample (58.6%) while significantly reducing the inference time (from 15.4s to 9.5s). This confirms that intelligent, task-aligned pre-filters can retain most of our framework's benefits at a fraction of the computational cost.

Table 7: **UG-Search Efficiency with Pre-filtering.** UG-Search is applied to smaller subset of visual candidates proposed by YOLOv8 and SAM2.

| | $V^*$ | Infer. Time |
|---|---|---|
| LLaVA-OV-7B | 74.4 | 0.5s |
| w/ **UG-Search** | 81.2 | 14.6s |
| + YOLOv8 Pre-filter | 74.9 | **9.0s** |
| + SAM2 Pre-filter | 78.2 | 26.5s |

Table 8: **UG-Sample Efficiency with Pre-filtering.** Top-32 frames are pre-filtered by SigLIP and DINOv2 which are then applied to UG-Sample.

| | V-MME | Infer. Time |
|---|---|---|
| LLaVA-OV-7B | 53.9 | 0.4s |
| w/ **UG-Sample** | 58.6 | 15.4s |
| + DINOv2 Pre-filter | 56.4 | **8.6s** |
| + SigLIP Pre-filter | 57.3 | 9.5s |

## 5 CONCLUSION

In this work, we addressed the challenge of fine-grained perception in MLLMs by demonstrating that a model's intrinsic uncertainty can serve as a powerful, proactive guide for localizing relevant information. Our Uncertainty-Guided (UG) framework provides a simple, unified, and training-free mechanism that solves three distinct tasks through uncertainty minimization, including Visual Search, Long Video Understanding, and Temporal Grounding. Our comprehensive experiments show that this approach enables standard MLLMs to achieve performance competitive with complex and specialized fine-tuned methods. This work validates harnessing a model's own uncertainty as an effective strategy for developing more capable multimodal models on complex visual tasks.

REPRODUCIBILITY STATEMENT

To ensure the reproducibility of our work, we will publicly release all source code for our Uncertainty-Guided (UG) framework and the scripts used for our experiments. All experiments were conducted using publicly available models and standard benchmarks, as detailed in Section 4. Comprehensive implementation details, including the specific hyperparameters for each task (e.g., crop size, window length, stride), decoding settings, and model versions, are provided in the main text (Section 4) and further elaborated in the Appendix (Section F). We adopted the publicly available LMMs-Eval Library (Zhang et al., 2024a) to standardize our evaluation protocol, ensuring that our results can be consistently reproduced. To improve clarity, the manuscript was polished for grammar and style using a large language model, with all final text reviewed and validated by the authors.

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

## A  INTERPRETATION OF UNCERTAINTY IN MLLMS

Quantifying uncertainty is a cornerstone of building reliable and interpretable machine learning systems (Ghahramani, 2015; Gal & Ghahramani, 2016; Lakshminarayanan et al., 2017). Entropy, a foundational concept from information theory (Shannon, 1948), provides a formal measure of the uncertainty inherent in a probability distribution. In the context of machine learning, entropy has long been integral to various algorithms and objective functions. For instance, it underpins the construction of decision trees and serves as the basis for the cross-entropy loss function, a standard for training deep neural networks in classification tasks (Goodfellow et al., 2016). These applications leverage entropy to assess prediction confidence, guide model optimization, and provide insights into model behavior.

The advent of large language models (LLMs) (Devlin et al., 2019; Brown et al., 2020; Touvron et al., 2023) has renewed interest in entropy as a vital tool for understanding and refining model performance (Shannon, 1951; Kadavath et al., 2022). Pretrained on massive text corpora, LLMs learn to generate reliable probability distributions over a predefined vocabulary. This capability allows for the direct application of entropy-based methods to address critical challenges such as hallucinations, where a model generates factually incorrect or nonsensical text (Azaria & Mitchell, 2023; Kuhn et al., 2023; Chen et al., 2024b). Recent studies have empirically demonstrated that incorrectly generated tokens often exhibit higher entropy than correct ones, establishing entropy as a practical proxy for prediction reliability (Xiao & Wang, 2021; Kadavath et al., 2022; Chen et al., 2024b).

Multimodal large language models (MLLMs) (Li et al., 2023a; Liu et al., 2023; Bai et al., 2023; Li et al., 2024; Chen et al., 2024d) extend the capabilities of LLMs by integrating visual encoders which process images into visual tokens that are subsequently fed into the model alongside text tokens. While this fusion of modalities enables remarkable new capabilities, MLLMs inherit many of the same uncertainty characteristics as their unimodal predecessors. Consequently, they exhibit similar entropy patterns, where higher entropy in the output distribution often correlates with incorrect or hallucinatory predictions (Leng et al., 2024; Zou et al., 2025; Wu et al., 2025b; Wang et al., 2024). This parallel suggests that uncertainty estimation via entropy can be similarly leveraged to enhance the interpretability of MLLM outputs.

While prior work has primarily used entropy for post-hoc hallucination detection, its potential as a proactive signal to guide a model's perceptual process, particularly for complex visual tasks, remains largely underexplored. Therefore, we propose a novel, training-free framework that leverages entropy to enhance MLLM performance on complex visual tasks. Our approach uses entropy not merely as an error detector but as an active guide for decision-making.

## B  THE INVERSE CORRELATION BETWEEN PERFORMANCE AND ENTROPY

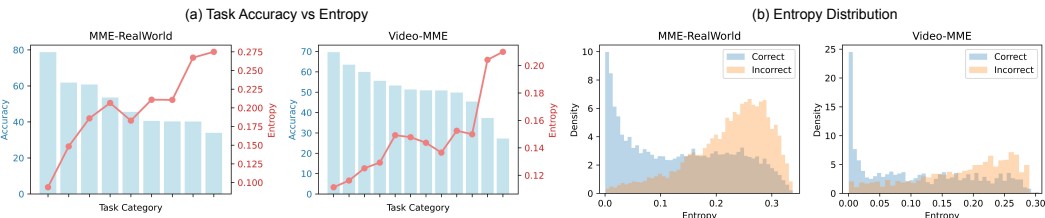

Figure 5: (a) Correlation between sub-task accuracy and entropy. (b) Entropy distribution for correct and incorrect predictions.

In this section, we systematically analyze the relationship between MLLM performance and output entropy on demanding multimodal tasks. While prior work has established that incorrectly generated tokens in language models often exhibit higher entropy, this phenomenon has not been thoroughly examined in the context of modern multimodal tasks that require fine-grained understanding of sparse information within noisy visual contexts. We begin by evaluating LLaVA-OneVision 7B (Li et al., 2024) on two challenging benchmarks: Video-MME (Fu et al., 2025) and MME-

RealWorld (Zhang et al., 2025c). MME-RealWorld assesses MLLMs on real-world scenarios across nine categories using 2k-resolution images, while Video-MME features a diverse set of 12 tasks on long-form videos averaging 17 minutes.

As shown in Figure 1 (a), we plot the accuracy for each sub-task in descending order alongside its corresponding average entropy. In both benchmarks, a clear inverse correlation emerges: sub-tasks with higher accuracy exhibit lower entropy, whereas more difficult sub-tasks yield higher entropy. We also calculate the Pearson Correlation Coefficient (Kirch, 2008) between entropy and accuracy, achieving -0.90 and -0.95 for MME-RealWorld and Video-MME respectively, which represents strong negative correlation. This finding aligns with principles from curriculum learning, where task difficulty can be measured by model uncertainty (Bengio et al., 2009; Kumar et al., 2010).

Furthermore, Figure 1 (b) depicts the entropy distribution for correct versus incorrect predictions. The entropy values for correct predictions are concentrated in a much lower range than those for incorrect ones. This observation confirms that findings from prior work on unimodal hallucination (Leng et al., 2024; Zou et al., 2025; Wu et al., 2025b; Wang et al., 2024), where incorrect outputs correlate with higher entropy, also hold for complex multimodal tasks.

## C    THEORETICAL DISCUSSION OF UG METHODS

In our paper, we hypothesize that the output entropy of a generative MLLM serves as a reliable proxy for its predictive uncertainty. Consequently, actively minimizing this entropy at inference time acts as a form of risk minimization for the downstream task.

Our approach's success rests on the principle of model calibration. While conventional neural networks are often imperfectly calibrated (Guo et al., 2017), our empirical results demonstrate that large-scale MLLMs retain a sufficient degree of calibration for entropy to be a strong indicator of correctness. Lower entropy signifies a more peaked, high-confidence distribution, which in a well-calibrated model, correlates directly with a higher likelihood of the prediction being correct.

From an information theory perspective, our framework can be viewed as an inference-time implementation of the Information Bottleneck principle (Tishby et al., 2000; Alemi et al., 2017), which posits that an optimal model should learn a compressed representation of the input that is maximally informative about the output. By searching for a visual input (e.g., a crop or frame) that minimizes the model's output entropy, our Uncertainty-Guided (UG) framework is effectively searching for the input that allows the model to form the most informative and least ambiguous internal representation with respect to the query. This low-entropy state signals that the model has successfully filtered out irrelevant information and isolated the data needed to solve the task.

We note that this relationship does not constitute a formal guarantee, as the precise mapping from entropy to accuracy is dependent on the model's architecture, training data, and the specific task distribution. However, our extensive experiments strongly validate this principle as a highly effective and generalizable heuristic for improving MLLM performance in a training-free manner.

## D    DETAILS OF BENCHMARKS

In this section, we present the benchmarks evaluated in our paper and indicate, in parentheses, the corresponding table where each benchmark is assessed with UG-methods. To ensure fair and reproducible comparisons, all benchmarks were evaluated using `LMMs-Eval` (Zhang et al., 2024a). We adopted the library's default evaluation protocol and inference parameters for each benchmark, including temperature, maximum token count, and beam search settings. For further details on the evaluation setup, please refer to the official repository: `https://github.com/EvolvingLMMs-Lab/lmms-eval`.

### D.1    VISUAL SEARCH

**Visual Search Benchmarks**

- $V^*$ **Bench** (Wu & Xie, 2024) (Table 1 and 2): $V^*$ Bench consists of 191 questions across two tasks: Attribute Recognition (115 samples) and Spatial Relationship Reasoning (76 samples). It utilizes high-resolution (approx. 2k), visually crowded images, focusing on fine-grained attribute recognition and spatial reasoning. The questions are multiple-choice; attribute tasks query object properties (color, material), while spatial tasks query relative positions. Each question provides 4 options for attribute tasks and 2 for spatial tasks. The predicted option is extracted from the generated output via a rule-based function and matched with the ground truth.

- **HR-Bench** (Wang et al., 2025a) (Table 1 and 2): HR-Bench includes ultra-high-resolution images (4k and 8K) and consists of two splits (HR-Bench 4K and HR-Bench 8k), containing 800 examples each (1,600 total). Each split is divided into two subtasks: Fine-grained Single-instance Perception (FSP) and Fine-grained Cross-instance Perception (FCP). It tests MLLMs on detailed perception and reasoning tasks, such as attribute recognition, OCR, and spatial relationships. Questions follow a multiple-choice format (A–D). The output string is matched with the ground truth by querying OpenAI gpt-3.5-turbo.

- **MME-RealWorld** (Zhang et al., 2025c) (Table 11): MME-RealWorld is a large-scale benchmark with 13,366 high-resolution images (approx. 2k) and 29,429 QA pairs, covering 43 subtasks across five real-world domains (Monitoring, OCR, Diagram/Table, Autonomous Driving, and Remote Sensing). Questions are multiple-choice (A–E). Predictions are extracted via rule-based patterns and matched with the ground truth.

**Standard VQA Benchmarks**

- **DocVQA** (Mathew et al., 2021) (Table 12): DocVQA evaluates document understanding, containing approximately 50,000 questions over scanned documents, invoices, and forms. It assesses the ability to extract and reason over textual and structural information. The output format consists of short text answers, often requiring exact string extraction.

- **POPE** (Li et al., 2023b) (Table 12): POPE focuses on hallucination detection. It includes around 3,000 image-question pairs designed to test whether the model hallucinates objects not present in the image. Questions are binary (Yes/No), measuring factual grounding.

- **TextVQA** (Singh et al., 2019) (Table 12): TextVQA evaluates Optical Character Recognition (OCR) and reasoning over text in natural images. It consists of 45,336 questions across 28,408 images. The benchmark tests reading comprehension in visual contexts (signs, labels). Outputs are short text answers, typically spanning one or two words.

## D.2   VIDEO FRAME SAMPLING

**Long Video Understanding Benchmarks**

- **Video-MME** (Fu et al., 2025) (Table 4, 3, and 13): Video-MME evaluates MLLMs on comprehensive video understanding. It consists of 900 videos (254 hours total) with 2,700 annotated QA pairs. Videos span six domains (e.g., knowledge, film, sports) and vary in duration from short (11s–2min) to long (30–60min). It integrates frames, subtitles, and audio to assess cross-modal understanding. Questions are multiple-choice, and predictions are extracted via rule-based parsing.

- **MLVU** (Zhou et al., 2025) (Table 4 and 3): MLVU targets videos ranging from 3 minutes to 2 hours. It consists of 3,102 questions (2,593 dev, 509 test) across diverse genres. Tasks include holistic understanding (summarization), single-detail reasoning (plot QA), and multi-detail reasoning (anomaly detection). It tests long-context reasoning and multi-task adaptability using a multiple-choice format.

- **LongVideoBench** (Wu et al., 2024) (Table 4 and 3): LongVideoBench focuses on long-context interleaved video-language understanding. It contains 3,763 videos with subtitles and 6,678 questions across 17 categories. Videos are up to one hour long, and questions follow a "referring reasoning" paradigm requiring retrieval of specific details.

**Short Video Benchmarks**

- **EgoSchema** (Mangalam et al., 2023) (Table 13): EgoSchema evaluates egocentric video understanding and commonsense reasoning. It contains ∼5,000 short clips paired with multiple-choice questions. The benchmark assesses temporal reasoning and schema-based understanding, often requiring prediction of future actions.
- **NextQA** (Xiao et al., 2021) (Table 13): NextQA focuses on next-event prediction and temporal reasoning in short videos. It includes ∼51,000 questions over 5,400 videos. Questions (multiple-choice and open-ended) test causal and temporal relationship inference.

### D.3 TEMPORAL GROUNDING

- **Charades-STA** (Gao et al., 2017) (Table 5): Charades-STA benchmarks temporal activity localization. It contains 12,408 clips with 16,128 sentence annotations. The model must predict start and end timestamps for a described activity. Performance is measured by computing the IoU score between the predicted timeline and the ground truth.
- **ActivityNet Captions** (Krishna et al., 2017) (Table 5): ActivityNet Captions focuses on dense video captioning and grounding. It includes 10,000 videos with ∼37,400 queries. The benchmark tests the alignment of multiple natural language descriptions with corresponding temporal segments in long videos.

## E DETAILS OF BASELINE METHODS

In this section, we detail the baseline methods compared to UG methods. We begin with general-purpose MLLMs used in our main experiments.

**General MLLM Baselines**

- **LLaVA-OneVision** (Li et al., 2024): An open-source model for unified image and video understanding, connecting a SigLIP encoder with Qwen2 via an MLP projector. Trained on large-scale instruction datasets with AnyRes, it supports scales of 0.5B, 7B, and 72B parameters.
- **Qwen2.5-VL** (Bai et al., 2025): A vision-language model trained on 4.1T tokens, supporting dynamic resolution and temporal scaling. It excels in object grounding, OCR, and long video understanding. Available in 3B, 7B, 32B, and 72B sizes.
- **InternVL-2.5** (Chen et al., 2024c): Built on the InternViT encoder and InternLM2.5/Qwen2.5, this model supports multi-image reasoning and video QA. It employs progressive scaling and dynamic high-resolution strategies (1B to 78B parameters).
- **LLaVA-Video** (Zhang et al., 2024b): An extension of LLaVA-OneVision fine-tuned on LLaVA-Video-178K. It unifies visual representation for images and videos, enabling temporal reasoning (7B and 72B parameters).
- **InternVideo-2.5** (Wang et al., 2025b): A video-centric MLLM built on InternVL-2.5, enhanced with Long and Rich Context (LRC) modeling. It introduces hierarchical token compression for videos up to 6× longer than previous versions (up to 8B parameters).

The following sections describe task-specific models (training-free or fine-tuned) compared in our paper.

### E.1 VISUAL SEARCH

**Training-free Methods**

- **ZoomEye** (Shen et al., 2025) (Table 2): A training-free method that represents images as a tree hierarchy. It uses tree search to dynamically zoom into relevant regions. While effective for high-resolution images, its applicability to standard VQA or video tasks remains unproven.

- **DyFo** (Li et al., 2025) (Table 1): DyFo employs a dynamic focus mechanism using MCTS and a visual expert to iteratively zoom into regions. It reduces hallucinations but relies heavily on external models, limiting its application in video and grounding tasks.

- **Rel-att** (Zhang et al., 2025a) (Table 2 and 11): Exploits internal attention/gradient maps to identify and crop relevant regions. While it boosts fine-grained task performance, improvements are marginal on recent MLLMs (e.g., Qwen2.5-VL), and video application is unexplored.

- **TextCoT** (Luan et al., 2024) (Table 2 and 11): A Chain-of-Thought framework for text-rich images, using a three-stage process (overview, coarse localization, fine-grained observation). Its application is primarily limited to text recognition tasks.

**Fine-tuned Methods**

- **SEAL** (Wu & Xie, 2024) (Table 1): Introduces MLLM-guided visual search to build a visual working memory. Unlike UG-methods, SEAL requires fine-tuning the MLLM to optimize the interaction between search and reasoning.

- **DeepEyes** (Zheng et al., 2025) (Table 1): Uses reinforcement learning (GRPO) to incentivize visual reasoning without supervised fine-tuning. While effective for perception, GRPO is GPU-intensive to train, limiting scalability to the video domain.

- **Thyme** (Zhang et al., 2025b) (Table 2 and 11): Enables MLLMs to perform image manipulations via code. It uses a two-stage training strategy (SFT followed by RL). It improves high-resolution perception but requires complex training pipelines.

## E.2 VIDEO FRAME SAMPLING

**VLM-based Methods**

- **BOLT** (Liu et al., 2025) (Table 3): A training-free method using query-guided inverse transform sampling. It relies on CLIP or SigLIP for similarity scores, which inherently lack fine-grained understanding.

- **AKS** (Tang et al., 2025) (Table 3): Adaptive Keyframe Sampling optimizes relevance and coverage within a token budget. It uses CLIP or BLIP for relevance, which limits fine-grained perception capabilities.

**Graph-based Method**

- **KFC** (Fang et al., 2025) (Table 3): Selects keyframes by optimizing relevance and diversity, then threads them with textual narratives from a lightweight captioner. This avoids fine-tuning but adds architectural complexity.

**MLLM-based Methods**

- **FRAG** (Huang et al., 2025) (Table 3): Independently scores frames for relevance and selects the Top-K. While it requires no fine-tuning, FRAG incurs high inference latency not fully addressed in the original paper.

- **Frame-Voyager** (Yu et al., 2024) (Table 4): Reframes selection as a ranking problem, training a module to predict high-reward frame sets. It requires independent fine-tuning for every new downstream task.

### E.3 TEMPORAL GROUNDING

**Training-Free Methods**

- **VTG-GPT** (Xu et al., 2024) (Table 5): A zero-shot method that debiases queries and converts frames to captions, using a proposal pipeline to predict timestamps.
- **TFVTG** (Zheng et al., 2024) (Table 5): Uses LLMs to decompose queries into sub-events and VLMs to score relevance, integrating results based on temporal logic.
- **TAG** (Lee et al., 2025) (Table 5): Incorporates temporal pooling, coherence clustering, and similarity adjustment to address semantic fragmentation in zero-shot grounding.

**Fine-Tuned Methods**

- **VTimeLLM** (Huang et al., 2024) (Table 5): Uses a three-stage training pipeline (alignment, boundary learning, instruction tuning). It requires extensive fine-tuning, hindering transferability.
- **TimeChat** (Ren et al., 2024) (Table 5): Features a timestamp-aware frame encoder and a sliding Q-Former. While accurate for timestamps, its utility for general VQA or long video tasks is unverified.
- **TimeMarker** (Chen et al., 2024a) (Table 5): Interleaves "Temporal Separator Tokens" with frame tokens. The introduction of custom tokens prevents easy application to standard MLLMs.

## F IMPLEMENTATION DETAILS

This section provides a comprehensive overview of the implementation details for our Uncertainty-Guided (UG) framework. To ensure fair and reproducible comparisons, all methods were implemented using the LMMs-Eval (Zhang et al., 2024a). Consequently, we adopted the library's standard inference parameters for each benchmark, including temperature, maximum token count, and beam search settings. The same decoding configuration was used for both the uncertainty scoring phase (Token Entropy or BRC score) and the final answer generation.

A notable implementation detail concerns batch processing. While our uncertainty scoring method is inherently parallelizable as the score for each candidate visual input can be computed independently, we observed a degradation in MLLM performance when using a batch size greater than one. This is a known issue with several open-source MLLMs, where batch inference can lead to less reliable outputs. To ensure maximum accuracy and reproducibility, we therefore set the batch size to one for all experiments, including both scoring and final inference. We anticipate that the runtime of our framework could be substantially reduced with hardware support for true parallel processing or as MLLMs evolve to better support reliable batch inference.

### F.1 UG-SEARCH

For UG-Search, we employ a sliding window to generate candidate crops from the high-resolution input image. Based on our ablation study (Section 4.4), we determined the optimal crop size to be one-sixth of the image's smaller dimension. For each candidate, both the original full-resolution image and the resized crop are provided as input to the MLLM. In Section G, we empirically found that resizing the smaller crop back to the original image resolution allows the MLLM to perceive its details more effectively, leading to better performance.

The benchmarks used for this task, $V^*$ Bench (Wu & Xie, 2024) and HR-Bench (Wang et al., 2025a), feature multiple-choice questions. The model's output typically consists of a single answer token (e.g., "A") followed by an end-of-sequence token ($\langle EOS \rangle$). We calculate the average token entropy across all generated tokens, including both the answer and the $\langle EOS \rangle$ token.

### F.2 UG-SAMPLE

In UG-Sample, we treat each video frame as a candidate visual input and use token entropy to score its relevance to the query. Our analysis showed that a window size of a single frame was optimal,

as identifying discrete, highly relevant moments is more critical for long-video QA than analyzing longer, continuous actions. Following established protocols and to manage computational costs, we sample the top 8 frames from a pool of 256 uniformly sampled candidate frames. The evaluation for this task was conducted on multi-choice QA benchmarks, including Video-MME (Fu et al., 2025), MLVU (Zhou et al., 2025), and LongVideoBench (Wu et al., 2024). Similar to UG-Search, we compute the average entropy across all generated tokens.

---

**Algorithm 1** Modified Kadane's Algorithm for Maximum Subarray

---

1: **Input:** A list of BRC scores, $scores$.
2: **Output:** Indices start, end of maximum sum subarray
3: $max\_sum \leftarrow -\infty$
4: $current\_sum \leftarrow 0$
5: $start \leftarrow 0, end \leftarrow 0, temp\_start \leftarrow 0$
6: **for** $i = 0$ **to** $length(scores) - 1$ **do**
7:    **if** $current\_sum \leq 0$ **then**
8:       $current\_sum \leftarrow scores[i]$
9:       $temp\_start \leftarrow i$
10:    **else**
11:       $current\_sum \leftarrow current\_sum + scores[i]$
12:    **end if**
13:    **if** $current\_sum > max\_sum$ **then**
14:       $max\_sum \leftarrow current\_sum$
15:       $start \leftarrow temp\_start$
16:       $end \leftarrow i$
17:    **end if**
18: **end for**
19: **return** $start, end$

---

### F.3 UG-GROUND

For UG-Ground, we calculate a Binary Response Confidence (BRC) score for each temporal segment using a sliding window. We prompt the MLLM with a standardized yes/no question for each window:

> Given the action: ⟨target action⟩, is this action depicted in the video?
> A. yes
> B. no
> Answer with the option's letter from the given choices directly.

Here, ⟨target action⟩ is the query event. The BRC score is then calculated from the model's output probabilities for the tokens corresponding to "A" and "B". This process transforms the video into a sequence of confidence scores, reducing the temporal grounding task to a Maximum Subarray Sum problem. We solve this efficiently in linear time using a modified version of *Kadane's Algorithm* (Bentley, 1984), detailed in Algorithm 1. The algorithm's output (the start and end indices of the maximum sum subarray) is mapped back to timestamps in seconds to produce the final result.

Based on our ablation study, which highlighted the importance of perceiving continuous motion for action localization, we use a sliding window of 15 frames with a stride of 1. To manage computational load, videos from Charades-STA Gao et al. (2017) were sampled at 3 FPS, while videos from ActivityNet Captions Krishna et al. (2017) were sampled at 1 FPS. For baseline comparisons, we restricted the input for the base MLLMs to 64 frames considering their context length limitations.

## G EXTRA ABLATION STUDIES

This section provides further ablation studies to analyze key design choices within our UG framework and their impact on performance and efficiency.

Table 9: **Ablation on UG-Search Parameters.** The results show that performance peaks when selecting the top-4 crops and resizing the crop (R) before inference provides a performance benefit.

Table 10: **Ablation on UG-Ground Parameters.** The results indicate that window size is the more critical factor than stride. Performance is robust to larger strides, allowing for significant efficiency gains with minimal accuracy loss.

|  | $V^*$ Bench |
|---|---|
| InternVL2.5-8B | 71.7 |
| Num Crop=1 w/o R | 82.2 |
| Num Crop=1 w/ R | 83.3 |
| Num Crop=2 w/ R | 83.3 |
| Num Crop=3 w/ R | 83.8 |
| Num Crop=4 w/ R | **84.8** |
| Num Crop=5 w/ R | 82.7 |

|  | Charades-STA |
|---|---|
| InternVideo2.5-8B | 32.3 |
| Stride=1, Frame=7 | 45.5 |
| Stride=1, Frame=15 | **51.0** |
| Stride=3, Frame=15 | **51.0** |
| Stride=5, Frame=15 | 50.9 |
| Stride=7, Frame=15 | 50.6 |
| Stride=9, Frame=15 | 50.2 |
| Stride=11, Frame=15 | 49.7 |

## G.1 ABLATION ON DESIGN CHOICES FOR UG-SEARCH

We investigate two important hyperparameters for UG-Search: the number of low-entropy crops selected for the final inference (top-k approach) and the effect of resizing the selected crop. As shown in Table 9, performance on $V^*$ Bench steadily improves as we increase the number of selected crops from one to four, peaking at 84.8%. This suggests that incorporating a broader set of high-confidence regions can provide better visual coverage. However, performance begins to decline beyond k=4, likely due to the introduction of less relevant or redundant information.

We also find that resizing the selected crop back to the original image size resolution provides a notable 1.1% performance gain (from 82.2% to 83.3%) compared to using the non-resized version. While a top-4 approach yields the best results in this setting, the optimal 'k' can vary across different MLLMs. Therefore, to maintain a simple, robust, and generalizable framework, we adopt the top-1 selection with resizing for all main experiments.

## G.2 ABLATION ON STRIDE FOR UG-GROUND

We analyze the impact of the sliding window's stride on the performance and efficiency of UG-Ground. The stride determines the temporal density of our uncertainty scoring, creating a trade-off between localization precision and computational cost. The results in Table 10 reveal that while dense temporal sampling (stride=1) is effective, the window length is the more critical factor, with a 15-frame window achieving the best performance (51.0% mIoU).

Crucially, we observe that performance is highly robust to an increased stride. A stride of 3 achieves the same peak accuracy as a stride of 1, and performance degrades very gracefully even with much larger strides (e.g., a stride of 11 still achieves 49.7% mIoU). This finding is significant as it demonstrates that UG-Ground can be made substantially more efficient in practice by using a larger stride. In the main experiments, we select a stride of 1 and frame window size of 15 for an optimal temporal grounding performance.

## H EXTRA RESULTS OF UG FRAMEWORK

This section provides additional experimental results that further demonstrate the versatility and effectiveness of our UG framework.

## H.1 EXTRA RESULTS WITH UG-SEARCH

Table 11: **UG-Search on the MME-RealWorld Benchmark.** Our training-free method significantly boosts perception performance, achieving results competitive with the fine-tuned Thyme model.

| Model | MME-RealWorld | | |
| --- | --- | --- | --- |
| | Reason. | Percep. | Overall |
| QwenVL2.5-7B | 33.4 | 51.5 | 49.3 |
| + TextCoT | 33.7 | 53.3 | 50.9 |
| + rel-attn | 34.2 | 53.4 | 51.1 |
| + Thyme | **43.7** | **60.1** | **58.1** |
| + UG-Search | 35.1 | 58.5 | 55.7 |

Table 12: **UG-Search on Standard VQA Benchmarks.** Our method provides consistent performance gains across diverse VQA tasks. The value in subscript ($\Delta$) indicates the improvement over the baseline.

| | TextVQA | POPE | DocVQA |
| --- | --- | --- | --- |
| LLaVA-OV-7B | 73.8 | 88.4 | 87.1 |
| + UG-Search$_\Delta$ | $75.1_{1.3}$ | $89.7_{1.3}$ | $88.2_{0.9}$ |
| Qwen2.5-VL-7B | 76.1 | 86.3 | 93.1 |
| + UG-Search$_\Delta$ | $\mathbf{79.7}_{3.6}$ | $86.7_{0.4}$ | $\mathbf{94.2}_{0.9}$ |
| InternVL2.5-8B | 77.0 | 90.5 | 91.4 |
| + UG-Search$_\Delta$ | $77.7_{0.7}$ | $\mathbf{90.8}_{0.3}$ | $92.3_{0.9}$ |

**UG-Search on MME-RealWorld.** MME-RealWorld (Zhang et al., 2025c) is a recently introduced benchmark that evaluates multimodal models in practical scenarios, with a particular focus on perception and reasoning skills. As shown in Table 11, our training-free method provides a substantial boost to the model's perception score (+7.0%), bringing its overall performance to 55.7 (+6.4%). This result significantly surpasses other training-free methods like TextCoT (Luan et al., 2024) and Rel-att (Zhang et al., 2025a) and is highly competitive with the fine-tuned Thyme model (Zhang et al., 2025b). These findings underscore that our uncertainty-guided approach is a powerful and efficient method for enhancing MLLM performance in practical, real-world scenarios.

**UG-Search on Standard VQA Tasks.** While our main experiments focused on high-resolution image benchmarks, we also investigate the generalizability of UG-Search on standard VQA tasks that use normal-resolution images but still demand fine-grained perception. We test on three diverse benchmarks: POPE (Li et al., 2023b) for hallucination detection, TextVQA (Singh et al., 2019) for optical character recognition, and DocVQA (Mathew et al., 2021) for document understanding. Table 12 shows that UG-Search provides consistent gains across all three tasks and models. For example, Qwen2.5-VL-7B gains +3.6% on TextVQA and +0.9% on DocVQA, while LLaVA-OV-7B and InternVL2.5-8B also benefit with smaller yet steady improvements. These results confirm that UG-Search generalizes beyond high-resolution settings and effectively enhances perception-oriented VQA benchmarks without training.

## H.2 EXTRA RESULTS WITH UG-SAMPLE

Table 13: **Extra Results with UG-Sample on Video QA Benchmarks.** The results demonstrate the robustness of our method on Video-MME with subtitles and its versatility on short video benchmarks like EgoSchema and NextQA. The value in subscript ($\Delta$) indicates the performance change from the baseline (green for gains, red for losses).

| Model | Video-MME (w/ sub.) | | | | EgoSchema | NextQA |
| --- | --- | --- | --- | --- | --- | --- |
| | Overall | Short | Medium | Long | | |
| *Video Length* | *17min* | *1.3min* | *9min* | *41min* | *1.7min* | *0.8min* |
| LLaVA-OV-7B | 58.8 | 70.8 | 56.1 | 49.4 | 59.1 | 77.1 |
| + UG-Sample$_\Delta$ | $60.6_{1.8}$ | $72.7_{1.9}$ | $58.8_{2.7}$ | $50.3_{0.9}$ | $\mathbf{60.3}_{1.2}$ | $77.5_{0.4}$ |
| LLaVA-Video-7B | 67.6 | 71.9 | 64.9 | 66.0 | 49.7 | 75.6 |
| + UG-Sample$_\Delta$ | $69.7_{2.1}$ | $\mathbf{74.8}_{2.9}$ | $\mathbf{68.3}_{3.4}$ | $66.1_{0.1}$ | $51.2_{1.5}$ | $77.5_{2.9}$ |
| InternVL2.5-8B | 61.1 | 70.9 | 60.7 | 51.9 | 56.3 | 76.6 |
| + UG-Sample$_\Delta$ | $\mathbf{61.7}_{0.6}$ | $69.2_{1.7}$ | $62.1_{1.4}$ | $\mathbf{53.8}_{1.9}$ | $57.1_{0.8}$ | $77.3_{0.7}$ |
| InternVideo2.5-8B | 58.5 | 67.1 | 57.8 | 50.7 | 56.4 | 76.5 |
| + UG-Sample$_\Delta$ | $58.7_{0.2}$ | $68.2_{1.1}$ | $57.0_{0.8}$ | $50.9_{0.2}$ | $58.1_{1.7}$ | $\mathbf{78.2}_{1.7}$ |

To further validate the generality of our UG-Sample framework, we conduct two additional sets of experiments. First, we assess its robustness in the presence of linguistic cues by evaluating on Video-MME with subtitles enabled. Second, we test its applicability on the EgoSchema (Mangalam

et al., 2023) and NextQA (Xiao et al., 2021) benchmarks, which feature much shorter average video lengths.

**Robustness to Linguistic Cues.** As shown in Table 13, UG-Sample continues to provide consistent performance improvements on Video-MME even when subtitles are available. This demonstrates that our visually-driven, uncertainty-based sampling remains an effective strategy and is not made redundant by the presence of textual information. For instance, LLaVA-Video-7B's performance improves by +2.1% overall, with a notable gain of +3.4% in the medium-length video category.

**Generality Across Video Lengths.** The results on EgoSchema and NextQA confirm that UG-Sample is beneficial even for videos with shorter average durations. Our method improves the LLaVA-OneVision baseline by +1.2% on EgoSchema and +0.4% on NextQA. The performance gains are more modest compared to long-video benchmarks as uniform sampling is less likely to miss crucial moments in shorter video. These results underscore the versatility of our approach that UG-Sample consistently remains as an effective strategy regardless of video length.

# I    QUALITATIVE RESULTS

In addition to the examples provided in the main paper, this section presents further qualitative results for each of our three tasks in Figure 6, Figure 7, and Figure 8.

**Visual search** Figure 6 further demonstrates the ability of UG-Search to overcome the perceptual limitations of the baseline model in both spatial reasoning and attribute recognition. For relational queries, such as determining the relative position of two objects, the baseline model frequently fails. In contrast, our method correctly localizes the relevant objects (e.g., the soccer ball and the bench) enabling it to resolve the spatial relationship accurately. Similarly, for attribute-based questions like identifying the color of a woman's dress, UG-Search successfully focuses on the correct region and provides the right answer where the baseline is misled by other elements in the scene. These examples highlight how uncertainty-guided focus directly translates to more reliable, fine-grained perception. The last row indicates cases where both the baseline and UG-Search fail, specifically when the target objects are small and relatively far apart. In such cases, the fixed bounding box can capture one object but may miss the other, resulting in an incorrect answer.

**Video key frame sampling** The examples in Figure 7 illustrate how UG-Sample mitigates both perceptual and temporal reasoning errors. In a fine-grained visual query about a man's hair color, the baseline model provides an incorrect answer, likely due to a poor selection of frames. Our method, however, correctly identifies the key frames where the attribute is clearly visible (e.g., hair color of the smoking man). For temporal reasoning tasks, such as counting the number of lion cubs appearing over time, UG-Sample selects a more comprehensive set of frames, leading to an accurate count where the baseline underestimates the total. Certain challenging cases remain difficult for both methods, such as counting the streams crossed by a cat, which cannot be solved solely by spatial reasoning. Nevertheless, the overall trend demonstrates that our approach yields more reliable visual and temporal understanding than the baseline.

**Video temporal grounding.** Figure 8 provides a clear visualization of UG-Ground's prediction. The baseline model's predictions are often misaligned, either starting too late, ending too early, or drifting entirely off the target event. For example, in the first example asking about "noodles", the baseline predicts 9.2s–32.5s, which mostly misses the correct window of 23.5s–69.2s, while our method successfully recovers the full range. Similarly, in the "cleaning a motorcycle" example, the baseline's predicted segment is too long and poorly aligned. In contrast, our method's prediction, guided by the distinct peak in the BRC score sequence, aligns almost perfectly with the ground truth. This pattern holds across multiple examples (e.g., "lighting the pumpkin" and "washing hair in a salon"), demonstrating that our uncertainty-based scoring effectively transforms the complex task of temporal grounding into a robust maximum subarray problem, yielding consistently accurate localization.

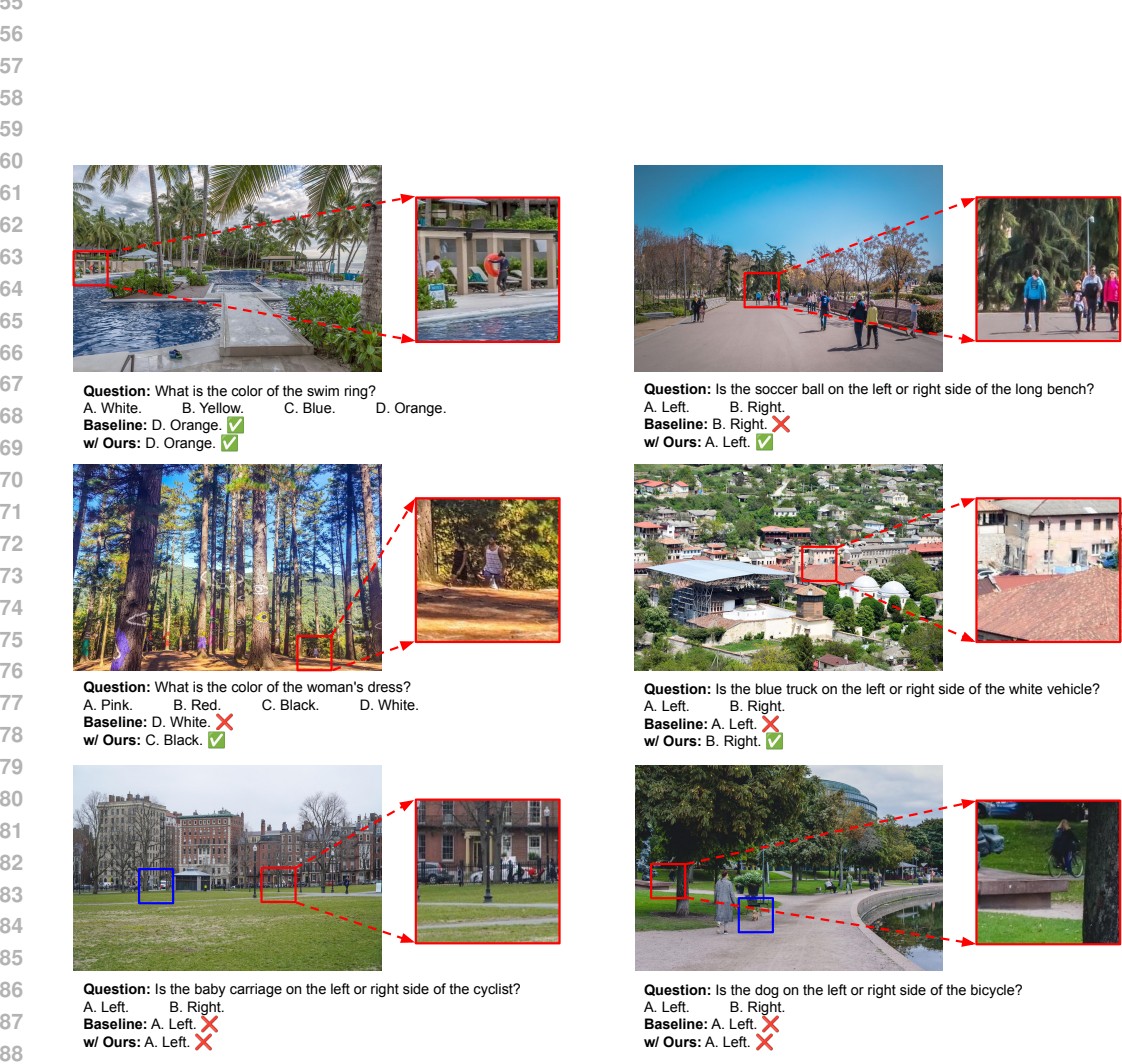

Figure 6: **Qualitative Results of UG-Search.** We compare qualitative results from UG-Search (ours) and InternV2.5-8B (baseline) on $V^*$ Bench. Red rectangle expresses the visual crop with the lowest entropy that is selected by UG-Search. UG-Search successfully capture the relevant object in examples of the first two rows while the last row exemplifies the failure cases where UG-Search capture one object and miss the another. Missed object is marked with blue rectangle.

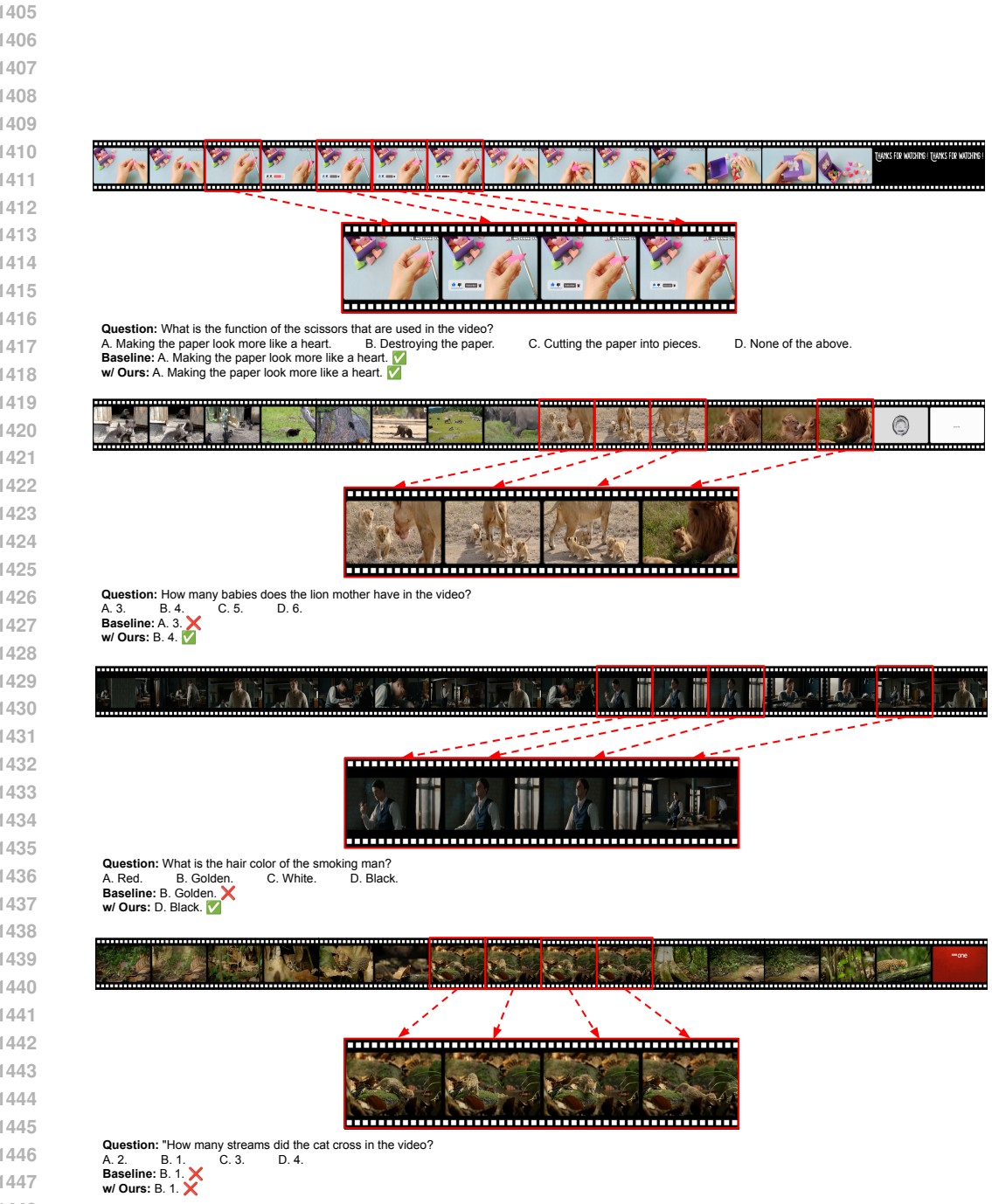

Figure 7: **Qualitative Results of UG-Sample.** We compare qualitative results from UG-Sample (ours) and InternV2.5-8B (baseline) on Video-MME. Red rectangle expresses the frames selected by UG-Sample. We provide a zoom-in version of the selected frames for better visualization. Selected frames are combined into a single context, and used for final inference to answer the query.

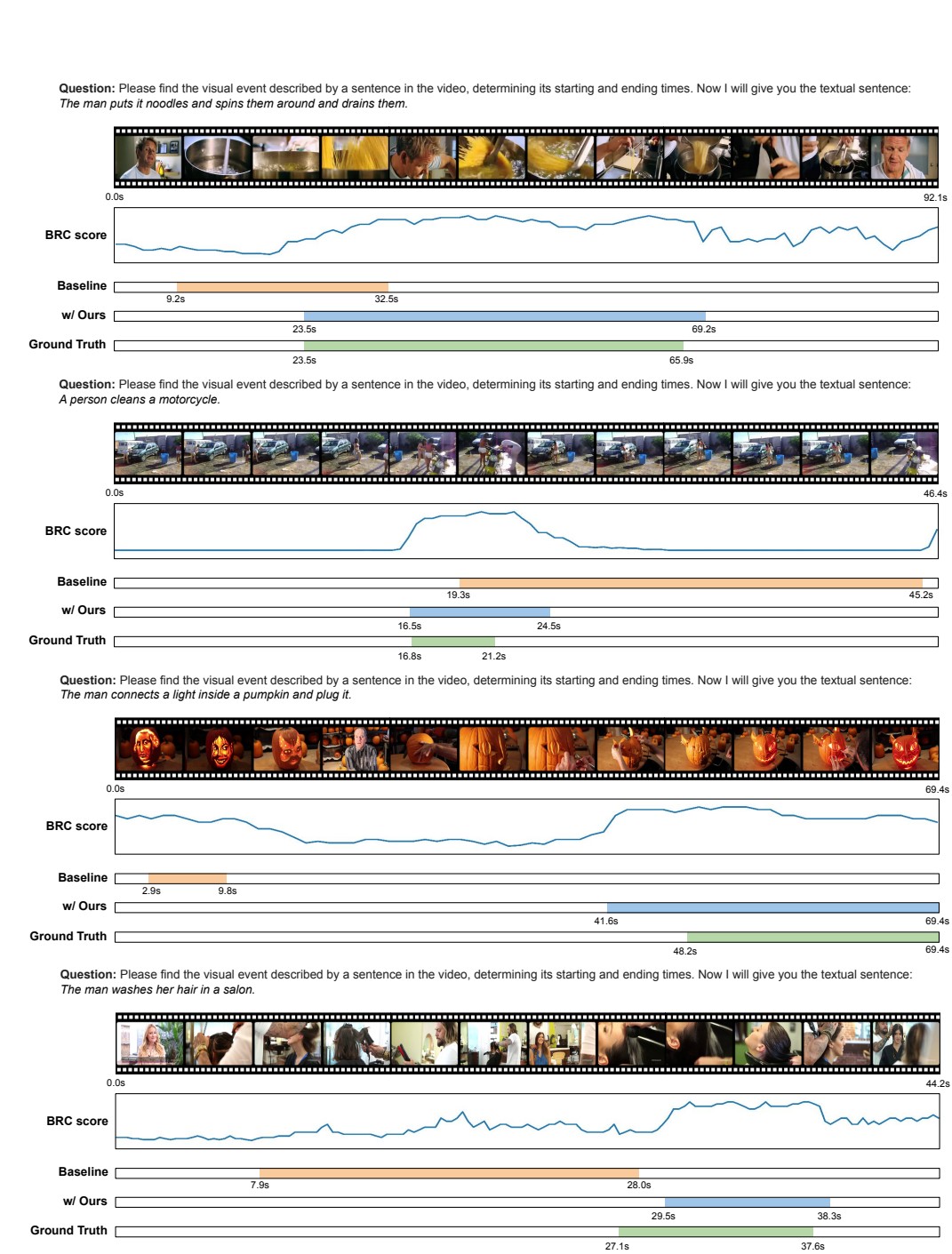

Figure 8: **Qualitative Results of UG-Ground.** We compare qualitative results from UG-Ground (ours) and InternV2.5-8B (baseline) on ActivityNet Captions. The orange, blue and green bars show the grounding results of baseline, UG-Ground, and ground truth respectively. Our UG-Ground method transforms the video into a sequence of BRC scores and then find the subarray with the maximum sum.

