# OpenReview forum: "Training-free Uncertainty Guidance for Complex Visual Tasks with MLLMs"
_ICLR.cc/2026/Conference — Submitted to ICLR 2026_

### Official Review · Reviewer_8dd3 · 2025-10-21

**Soundness:** 2
**Presentation:** 3
**Contribution:** 2
**Rating:** 4
**Confidence:** 5

**Summary:**

This paper introduces a training-free uncertainty guided framework for MLLMs to improve the fine-grained perception of visual information.  The proposed method estimates uncertainty from entropy-based measures derived from multiple forward passes with different portions of the visual input. It then selects the visual input with the lowest uncertainty as input for the final answer. The proposed method is applied to three tasks, including fine-grained visual search with high-resolution images, long video VQA, and temporal grounding. Experimental results show the effectiveness of the proposed method.

**Strengths:**

1. The idea of using uncertainty to select where to focus is interesting.
2. The authors have implemented the framework on different multimodal tasks to validate its effectiveness.
2. The overall writing is clear.

**Weaknesses:**

1. The evaluated benchmarks primarily involve tasks with short, discrete answers (e.g., single words or multiple-choice options). However, in realistic MLLM applications, outputs often take the form of longer sentences or paragraphs. The proposed entropy-based uncertainty estimation has not been validated in such open-ended generation settings, leaving its effectiveness for real-world scenarios unclear.

2. The method assumes that the type of question (e.g., yes/no, multiple choice, open-ended) is known beforehand to compute entropy appropriately. In practice, however, this information is rarely available during inference, which limits the applicability and automation of the proposed uncertainty estimation pipeline.

3. The approach always selects the top-1 region based on uncertainty. This assumption may fail when questions require reasoning over multiple fine-grained visual regions.


4. The method requires performing tens to hundreds of inference passes per sample to estimate uncertainty, which introduces substantial computational overhead. Compared with the base model or other lightweight techniques, the proposed approach is significantly less efficient, raising concerns about its practical scalability for large-scale or real-time applications.

5. Additional fine-grained benchmarks like MME-RealWorld should be added to include more diverse question types.

**Questions:**

1. What are the results on the two video-related tasks with the Qwen2.5-VL-7B baseline?
2. How would the proposed framework affect the performance of the general MLLM benchmarks that might not require very detailed fine-grained information?

---

> ### Author Response · Authors · 2025-11-23
> **Response to Reviewer 8dd3 (1/2)**
>
> We appreciate your insights. We strengthened the paper by adding runtime/performance trade-off results (revised Fig. 3 and Sec. 4.5) based on your comment. Clarifications for each concern is provided below.
>
> > **“1. The evaluated benchmarks primarily involve tasks with short, discrete answers…”**
>
> **Response:** We acknowledge that our evaluated benchmarks primarily involve short, discrete answers, as we followed the evaluation schemes used in prior work (e.g., ZoomEye [1] and Deepeye [2] for visual search; BOLT [3] and AKS [4] for video sampling), which focus on multiple-choice benchmarks for visual search and video sampling tasks. These benchmarks are relatively new, and to the best of our knowledge, there is currently **no widely adopted benchmark for fine-grained reasoning in long-form generation.**
>
> We believe that standard captioning benchmarks are not suitable for this purpose because they emphasize global image descriptions and rely on metrics such as ROUGE, BLEU, or external LLM-based similarity scoring, which do not directly measure fine-grained reasoning ability.
>
> Importantly, our entropy-based uncertainty estimation is **in principle applicable to open-ended generation**, as entropy is computed over all generated tokens regardless of response length. If the model is uncertain about specific tokens, this uncertainty will be reflected in the overall entropy score.
>
> That said, we agree that further benchmarking in open-ended settings is necessary to validate effectiveness for real-world scenarios involving long-form outputs. We view this as an **important direction for future work.**
>
>
>
> > **”2. The method assumes that the type of question (e.g., yes/no, multiple choice, open-ended) is known beforehand to compute entropy appropriately.”**
>
> **Response:** We would like to clarify that our method **does not require prior knowledge of the question type.** The entropy score is computed uniformly over all generated tokens, regardless of whether the question is yes/no, multiple-choice, or open-ended.
>
> To demonstrate this, we include evaluations on diverse benchmarks covering different formats (as shown in Tab. 12, Appendix):
>
> **- POPE** – Yes/No format
>
> **- TextVQA and DocVQA** – Open-ended short answers
>
>
> where **UG-Search consistently improves performance across formats.**
>
> Additionally, we newly evaluate **GQA** and **VQAv2** in general response, which primarily involve single-word or short-phrase answers. These results further confirm that entropy-based uncertainty estimation is **applicable across question types without requiring explicit format information.**
>
>
> > **“3. The approach always selects the top-1 region based on uncertainty.…”**
>
> **Response:** Thank you for raising this point about multi-region reasoning. For full details, we refer the reviewer to **4. Locality & Relational Reasoning** in the Global Author Response. Briefly, incorporating **Top-K crops** leads to consistent gains across models (e.g., InternVL2.5-8B improves from 83.3% to 84.8% with Top-4; Qwen2.5-VL-7B reaches 81.7% with Top-5). However, larger K introduces two limitations: degradation in some MLLMs and increased inference cost. Therefore, while multi-crop approaches are effective in certain cases, Top-1 remains the most reliable and computationally efficient default. Qualitative failure examples are provided in Appendix I.
>
> > **“4. The method requires performing tens to hundreds of inference passes.…”**
>
> **Response:** Thank you for raising this issue. We acknowledge the computational overhead and direct the reviewer to **1. Computational Overhead** for full quantitative analysis and new ablations. In short, our design aligns with **inference-time scaling**, where additional compute enables significantly stronger grounding performance. Notably, UG’s scoring passes are **inherently parallel**, allowing large reductions in wall-clock runtime when batched or distributed, an efficiency advantage absent in sequential reasoning pipelines.
>
> Furthermore, **Fig. 3** and **Sec. 4.5** newly introduced strategies that substantially improve efficiency, including adjustable input granularity, decoupled lightweight scorers, external filtering, and temporal stride adjustment. Therefore, while additional compute is required, UG provides **flexible and controllable runtime** rather than fixed overhead, supporting practical deployment.
>
> [1] Shen, H. et. al. (2025). Zoomeye: Enhancing multimodal llms with human-like zooming capabilities through tree-based image exploration. In EMNLP 2025.
>
> [2] Zheng, Z. et. al. (2025). DeepEyes: Incentivizing" Thinking with Images" via Reinforcement Learning. arXiv preprint arXiv:2505.14362.
>
> [3] Liu, S. et. al. (2025). BOLT: Boost Large Vision-Language Model Without Training for Long-form Video Understanding. In CVPR 2025.
>
> [4] Tang, X. et. al. (2025). Adaptive keyframe sampling for long video understanding. In CVPR 2025.

---

> > ### Author Response · Authors · 2025-11-23
> > **Response to Reviewer 8dd3 (2/2)**
> >
> > > **“5. Additional fine-grained benchmarks like MME-RealWorld should be added to include more diverse question types.”**
> >
> > **Response:** We would like to clarify that **MME-RealWorld** results are already included in **Appendix Tab. 11.**
> >
> >
> > > **”Question: What are the results on the two video-related tasks with the Qwen2.5-VL-7B baseline?”**
> >
> > **Response:** Below are the results for **Qwen2.5-VL-7B** on the two video-related tasks, video sampling and temporal grounding, using the same experimental settings as in the main paper. Our UG-framework consistently outperforms the Qwen2.5-VL-7B baseline on both tasks, demonstrating its effectiveness across different MLLMs.
> >
> >
> > **Table 9. Qwen2.5-VL-7B with UG-Sample**
> > | **Model**          | **Videomme Overall** | **Short** | **Medium** | **Long** | **MLVU** | **LVB** |
> > |---------------------|----------------------|-----------|------------|----------|----------|---------|
> > | Qwen2.5-VL-7B       | 53.7                | 62.7      | 51.7       | 46.8     | 53.9     | 52.7    |
> > | w/ UG-Sample       | 59.8                | 69.4      | 58.9       | 51.0     | 55.3     | 60.5    |
> >
> > **Table 10. Qwen2.5-VL 7B with UG-Ground**
> > | **Model**        | **Charades-STA R@0.3** | **R@0.5** | **R@0.7** | **mIoU** | **ActivityNet R@0.3** | **R@0.5** | **R@0.7** | **mIoU** |
> > |-------------------|-------------------------|-----------|-----------|----------|-------------------------|-----------|-----------|----------|
> > | Qwen2.5-VL-7B           | 61.1                    | 43.7      | 22.9      | 41.1     | 19.0                    | 9.5       | 4.1       | 14.3     |
> > | w/ UG-Ground     | 70.1                    | 51.0      | 30.1      | 48.4     | 54.4                    | 31.4      | 16.5      | 37.9     |
> >
> >
> >
> > > **”Question: How would the proposed framework affect the performance of the general MLLM benchmarks that might not require very detailed fine-grained information?”**
> >
> >
> > **Response**: We appreciate the question and refer the reviewer to **3. Baselines & Generalization** of the Global Author Response. In summary, new experiments on **GQA** and **VQAv2** (Table 3 in Global Author Response) show that UG-Search preserves or improves performance on general MLLM benchmarks, indicating no degradation when fine-grained reasoning is not required. Additional results on **TextVQA, POPE**, **DocVQA** (Tab. 12 of main paper) and short-video benchmarks **Egoschema** and **NextQA** (Tab. 13 of main paper) further support the general applicability of our method.

---

### Official Review · Reviewer_Vcnq · 2025-10-31

**Soundness:** 2
**Presentation:** 3
**Contribution:** 3
**Rating:** 2
**Confidence:** 4

**Summary:**

This paper proposes UG, a training-free uncertainty-guided framework that improves fine-grained perception for MLLMs. The key observation is that when the model attends to the most relevant visual region, its output uncertainty decreases. Leveraging this, UG uses the MLLM itself to score candidate image crops / video frames / temporal windows by uncertainty, selects the most informative ones, and then performs the final inference only on these selected regions. Experiments on high-resolution image QA, long-video QA, and temporal grounding show that UG achieves strong gains over prior methods, including several finetuned systems, without any model training. The paper argues that this demonstrates uncertainty as an effective guidance signal for multimodal reasoning.

**Strengths:**

- **Simple and general training-free approach**: The method requires no model updates and can be directly applied to existing MLLMs without additional training or supervision.

- **Strong empirical gains**: Demonstrates substantial improvements across high-resolution image QA, long-video understanding, and temporal grounding benchmarks.

- **Model-agnostic**: Works with multiple popular multimodal models, showing broad applicability and compatibility across architectures.

- **Practical for real use cases**: Particularly valuable in scenarios where retraining is infeasible.

**Weaknesses:**

- **Inference overhead**: The approach incurs substantial inference-time cost, since it requires running the MLLM on many candidate crops/frames/windows to compute uncertainty before the final prediction. This runtime overhead is significantly higher than naive baselines and also higher than prior methods that rely on lightweight selectors or external scorers, making scalability a concern for long videos and high‐resolution inputs.

- **Core motivation not fully validated**: The key hypothesis that relevant regions exhibit lower uncertainty has only been evaluated when zooming toward the correct location. The paper does not systematically analyze *mislocalized* or distractor regions, so the claim that uncertainty reliably guides search remains primarily empirical.

- **Inconsistent uncertainty formulation**: The method uses token entropy for image and video frame selection but switches to a yes/no logit margin (BRC) for temporal grounding. The latter behaves more like a decision confidence score rather than a general uncertainty estimate. A calibration or ablation comparing these metrics would strengthen conceptual consistency.

- **Locality assumption and relational limits**: While the approach improves some relational cases, the appendix also presents failure examples where relevant objects are spatially distant, suggesting the method fundamentally relies on localizable evidence. A more systematic analysis of relational scenes and potential multi-patch extensions would clarify this limitation.

- **Compute fairness perspective not fully addressed**: Some strong prior systems amortize selection cost via training lightweight selectors or using external models, whereas this method shifts all cost to inference.

- **Task description could be more self-contained**: While standard benchmarks are used, the manuscript would benefit from briefly summarizing the problem format and evaluation protocol for each task/dataset to aid readers unfamiliar with these benchmarks.

**Questions:**

- Please refer to the points listed under *Weaknesses*.
- I am also curious about the sensitivity of the uncertainty signal to prompt phrasing. Have you tried alternative templates or paraphrased prompts when computing entropy scores, and does the performance of UG remain stable under such variations?

---

> ### Author Response · Authors · 2025-11-23
> **Response to Reviewer Vcnq (1/2)**
>
> Thank you for highlighting these important points. According to your review, we have updated the manuscript with more detailed efficiency analysis (new performance–runtime trade-offs in Fig. 3 and Sec. 4.5), improved treatment of uncertainty reasoning (updated Fig. 1 and Sec. 3.1), and expanded experimental documentation and baseline descriptions in Appendix Sections D–E. Our detailed responses follow.
>
> > **“1. Inference overhead…”** and **”5. Compute fairness perspective not fully addressed…”**
>
> **Response:** We acknowledge the inference-time overhead and we refer the reviewer to **1. Computational Overhead** of the Global Author Response for detailed quantitative comparisons and new ablation results. In summary, our approach follows the emerging paradigm of **inference-time scaling**, where additional test-time compute is exchanged for significantly stronger visual grounding. Crucially, unlike sequential CoT-style reasoning methods, UG’s uncertainty scoring is **inherently parallelizable**, enabling substantial wall-clock speed reductions when batching or distributed across GPUs, an advantage not available to autoregressive approaches.
>
> To be specific, we provide several mechanisms that substantially reduce runtime while maintaining accuracy, as summarized in **Fig. 3** and **Sec. 4.5**:
>
> - Tunable input granularity (Fig. 3) to control accuracy–latency trade-offs.
>
> - Decoupling scoring and answering (Sec. 4.5) via lightweight scorers.
>
> - External pre-filters (Sec. 4.5) to prune candidates early.
>
> - Stride-based temporal subsampling (in Global Author Response), shown in new ablation tables, which yields large speed gains on long videos.
>
> Together, these results demonstrate that UG offers **flexible and controllable inference cost**, substantially lower runtime when efficiently parallelized, and scalability suitable for high-resolution or long-duration inputs. For full results and extended tables, we kindly direct the reviewer to the Global Author Response.
>
>
> > **“2. Core motivation not fully validated…”**
>
> **Response:** Thank you for raising concern about our motivational study. We kindly refer the reviewer to **2. Hypothesis Validity** of the Global Author Response where we have provided new experimental analysis (replacing Fig. 1) demonstrating that crops containing the target object consistently exhibit lower entropy than "distractor" crops.
>
>
>
> > **3. “Inconsistent uncertainty formulation…”**
>
> **Response:** We agree that our method uses different uncertainty metrics for spatial selection (token entropy) and temporal grounding (BRC score). This choice is motivated by the nature of the tasks. For UG-Search/Sample, token entropy effectively identifies regions or frames with high uncertainty, guiding refinement toward informative content. For UG-Ground, the question format (“Is an *\<Query\>* event present?”) requires distinguishing between confident *yes* and confident *no* responses. **Entropy alone cannot make this distinction** because both cases exhibit low entropy. Therefore, we adopt the Binary Response Confidence (BRC) score, which measures the logit margin between “yes” and “no” options (see Lines 146–150).
>
> To address your suggestion, we conducted an **ablation comparing entropy vs. BRC score** within the UG-Sample framework on **LongVideoBench** as follows.:
>
> **- Entropy-based UG-Sample**: Average token entropy for prompts of the form
>  *"\<Frame Window\> \<Query\>"* (same as main paper).
>
> **- BRC-based UG-Sample**: Binary-choice prompt similar to UG-Ground (referring to Section F.3):
>  *"\<Frame Window\> Given the question \<Query\>, is this query relevant to the video clip? A. yes B. no. Answer with the option’s letter."*
>
> Both metrics were applied to 256 candidate frames, sampling 8 frames for LLaVA-OneVision-7B. Results:
>
> **Table 7. Performance on LongVideoBench**
> | Method                   | LongVideoBench  |
> |---------------------------------|------------------|
> | LLaVA-OneVision-7B         | 54.3            |
> | w/ BRC-based UG-Sample     | 58.0            |
> | w/ Entropy-based UG-Sample  | 59.5            |
>
> **Key Findings:**
>
> - Entropy-based UG-Sample achieves the highest improvement, likely because the additional prompt for BRC introduces noise that affects the model’s ability to extract key information.
>
> - Both methods outperform the baseline by a large margin, showing that BRC remains a reliable metric for identifying relevant frames.
>
> We also provide additional visualizations in Fig. 5 (Sec. B) (Entropy distribution for correct vs. incorrect answers.) and Sec. I (Qualitative examples of BRC scores across video timelines). These analyses illustrate how both measures behave and support their roles in guiding prediction.

---

> > ### Author Response · Authors · 2025-11-23
> > **Response to Reviewer Vcnq (2/2)**
> >
> > > **“4. Locality assumption and relational limits…”**
> >
> > **Response:** We appreciate the concern regarding multi-region reasoning. We kindly ask the reviewer to refer to **4. Locality & Relational Reasoning** of the Global Author Response for detailed results. In summary, we extended our experiments with **Top-K crops**, showing consistent accuracy gains across models (e.g., InternVL2.5-8B improves from 83.3% → 84.8% with Top-4, and Qwen2.5-VL-7B reaches 81.7% with Top-5). However, handling multiple crops introduces challenges: some MLLMs degrade with many images and inference cost increases with higher K value. Thus, while multi-crop extensions are promising, Top-1 remains the most robust and efficient default.
> >
> >
> > > **“6. Task description could be more self-contained…”**
> >
> > **Response:** We thank the reviewer for the comment regarding the task description. As detailed in **3. Baselines & Generalization** of the Global Author Response, we have significantly expanded the Appendix to include comprehensive descriptions of all datasets (**Section D**) and baseline methodologies (**Section E**).
> >
> >
> >
> > > **“Question: I am also curious about the sensitivity of the uncertainty signal to prompt phrasing. Have you tried alternative templates or paraphrased prompts when computing entropy scores, and does the performance of UG remain stable under such variations?”**
> >
> > **Response:** During the scoring stage for entropy calculation, we directly used the standard pre-prompts and post-prompts (e.g., Answer with the option's letter from the given choices directly.) provided by each benchmark along with the task query (e.g., what is the color of the truck?). While this setup ensures reproducibility and fair comparison, it is hard to systematically paraphrase the text prompt pre-defined by each dataset.
> >
> > Therefore, we conducted experiments with visual prompting inspired by [8]  where we added the red rectangle in the original image to denote crop region and updated the prompt accordingly. Specifically, we compared:
> >
> > **- Original template of UG-Search:**
> >
> > *\<original image\> \<crop\> \<Query\>*
> >
> > **-Alternative template (named *VPrompt*, explicitly highlighting the crop region)**:
> >
> > *\<original image with red rectangle\> \<crop\> First image is the main image, and the red rectangle marks the focused region shown in the second image. \<Query\>*
> >
> > We tested **LLaVA-OneVision-7B** on V* Bench across different crop sizes ($1/2$, $1/3$, and $1/6$ ) to assess robustness under varying visual granularity. The results in Table 8 shows that the performance differences between templates were minimal, indicating that the UG framework is largely robust to this prompt setting.
> >
> > **Table 8. Performance on V\* Bench**
> > | **Model**                                | **Attribute** | **Spatial** | **Overall** |
> > |-----------------------------------------|--------------|------------|------------|
> > | LLaVA-OV 7B                             | 79.1         | 67.1       | 74.4       |
> > | w/ UG-Search (1/2)                       | 85.2         | 75.0       | 81.2       |
> > | w/ UG-Search (1/2) VPrompt     | 84.4         | 73.7       | 80.1       |
> > | w/ UG-Search (1/3)                       | 92.2         | 69.7       | 83.3       |
> > | w/ UG-Search (1/3) VPrompt               | 92.2         | 69.7       | 83.3       |
> > | w/ UG-Search (1/6)                       | 94.8    | 75.0       | 86.9       |
> > | w/ UG-Search (1/6) VPrompt               | 95.7         | 73.7       | 86.9       |
> >
> >  [8] Shtedritski, A., Rupprecht, C., & Vedaldi, A. (2023). What does CLIP know about a red circle? Visual prompt engineering for VLMs. In ICCV 2023.

---

> > > ### Comment · Reviewer_Vcnq · 2025-11-28
> > >
> > > Thank you for the detailed rebuttal and the additional experiments. Many of the points I raised have been addressed, especially the analyses on distractor regions, the justification and ablation regarding uncertainty metrics, and the extended discussion on locality and multi-crop variants.
> > >
> > > I currently have no further questions. If the system later allows updating the score, I will consider adjusting my rating accordingly.

---

### Official Review · Reviewer_Db5z · 2025-11-01

**Soundness:** 3
**Presentation:** 3
**Contribution:** 3
**Rating:** 6
**Confidence:** 4

**Summary:**

This paper proposes a training-free test-time-scaling method to enhance fine-grained multimodal performance. For input high-resolution images or long videos, their method first conducts sampling and gains different visual clues and reasoning paths, then they select the final answers based on uncertainty or 'Yes-No' confidence. They demonstrate effectiveness on three tasks: Visual Search, Long Video Understanding, and Temporal Grounding.

**Strengths:**

1. The experiments are comprehensive. They demonstarte effectiveness on different tasks (image / video) and models (LLaVA, InternVL).
2. The motivation is reasonable. The Figure 1 is necessary for the story.

**Weaknesses:**

1. The method is a little simple, and the novelty is limited. They generate multiple answers with different viusal enhancement versions, then select final answers based on generation probability.
2. The time cost is little large. Especially, for long video understanding, they treat each frame (or short window) as visual candiate, but maybe some questions need motion across frames, enumerating all possible video clips would be computationally expensive. Similarly, for video temporal grounding tasks, a 'Yes-No' score must be calculated for every second.

**Questions:**

1. See weakness.
2. If the teaser image can also demonstrate uncertainty-video(clip) relationship, it will be better.

---

> ### Author Response · Authors · 2025-11-23
> **Response to Reviewer Db5z**
>
> We appreciate your valuable comments. According to your comments, we have revised the paper to include comprehensive runtime evaluations (updated Fig. 3 and Sec. 4.5). Now, we respond below.
>
> > **“1. The method is a little simple, and the novelty is limited…”**
>
> **Response:** We respectfully argue that **simplicity is a core strength** of our work, ensuring robustness and broad applicability, rather than a limitation. While the mechanism is straightforward, the concept, using intrinsic model uncertainty as a proxy for visual informativeness, is a novel contribution that has not been explored for visual granularity control.
>
> **(1) Novelty (Uncertainty as a Visual Sensor)**: The reviewer characterizes our method as merely "generating multiple answers." However, the core innovation is **repurposing token entropy**. Prior works use uncertainty primarily to reject bad answers (hallucination detection). We are the first to use it **proactively** to *discover* the most relevant visual inputs (crops, frames, or windows). This transforms uncertainty from a passive safety metric into an active selection signal.
>
> **(2) Effectiveness and Generalization**: Despite its simplicity, the UG-framework is surprisingly effective. Our method outperforms baselines across **13 diverse benchmarks** (Visual Search, Video Sampling, and Temporal Grounding) and across all visual input granularity and all MLLM scale settings (as shown in Fig. 3).
>
> **(3) Practical Utility**: Unlike complex, architecture-specific designs (e.g., specialized attention heads or custom adapters), our "simple" design enables the method to be:
>
> **-Training-Free & Plug-and-Play:** It can be immediately applied to any off-the-shelf MLLM (from 1B to 72B parameters) without fine-tuning.
>
> **-Robust:** It consistently improves performance where more complex, fragile pipelines often fail to generalize across different model families.
>
>
> > **“2. The time cost is little large…”**
>
> **Response:** We kindly refer the reviewer to the **1. Computational Overhead** of the Global Author Response for a detailed discussion addressing concerns about computational overhead. The Global Author Response is summarized as follows:
>
> **- Tunable input granularity (Fig. 3)** enabling explicit control over accuracy–latency trade-offs.
>
> **- Decoupling scoring from answering (Sec. 4.5)** using smaller scorers.
>
> **- External pre-filters (Sec. 4.5)** to prune search space.
>
> **- Stride-based temporal subsampling (in Global Author Response)** that significantly reduces inference time while preserving accuracy on video-based tasks.
>
> These results show that UG methods offer **flexible and controllable inference-time cost**, and the overhead can be substantially reduced via batching, distributed processing, or the design choices above. For full quantitative evidence and updated ablations, please see the Global Author Response.
>
>
> > **“Question: If the teaser image can also demonstrate uncertainty-video(clip) relationship, it will be better.”**
>
> **Response:** We have updated Fig. 1 of the main paper, based on the reviews we received, regarding our key hypothesis. Now, it demonstrates that crops containing the target object consistently exhibit lower entropy than "distractor" crops. We kindly ask reviewer to refer to our updated paper and **2. Hypothesis Validity** of the Global Author Response for details.

---

### Official Review · Reviewer_sqRq · 2025-11-01

**Soundness:** 2
**Presentation:** 2
**Contribution:** 2
**Rating:** 2
**Confidence:** 4

**Summary:**

This paper proposes a training-free method to enhance the ability of MLLMs to focus on small regions on the image or video when answering questions. The proposed method divides the input image or video into segments, computes the MLLM’s answer uncertainty on each segment, and uses the uncertainty scores to select the segments that minimize the MLLM’s uncertainty. The paper provides several experiments to show the effectiveness of the proposed method in answering questions about small objects on images, questions on long videos, and localizing events in videos.

**Strengths:**

The direction of training-free approaches to enhance MLLMs' visual performance seems interesting and valuable to me. The proposed method is simple and architecture-agnostic, so it is easy to apply to various MLLMs. The method seems effective in answering multi-choice questions about small objects and localizing events in videos, so it can find application in these settings, although its broader effect on general visual question answering and video understanding performance remains unclear.

**Weaknesses:**

1. The evidence provided in Figure 1 does not support the paper’s main hypothesis that “an MLLM’s intrinsic uncertainty can be actively minimized at inference time to guide it toward the correct answer” as claimed in Lines 106-133. The results in Figure 1 suggest that zooming-in on the correct image region will minimize the model’s uncertainty, but not the other way around which is the paper’s hypothesis (that minimizing the model’s uncertainty will zoom-in on the correct image region). Note that there are many examples that contradict the paper’s hypothesis, for example: Imagine the question “Is there any dog in this image?” on a picture of a dog, then searching for the image region that minimizes the model’s uncertainty could focus on a completely empty background region where the model can be very certain that there is no dog! The line of research on adversarial attacks provides various such examples for unreliability of optimizing softmax-induced uncertainty. The paper must provide relevant experiments to support its key hypothesis, or narrow it down.

2. The choices of hyper-parameters (size of crop, temporal segment, k in top-k) seem to be made on the test videos (per section 4.4), and are therefore are likely overfitting to the small test datasets. The correct approach is to pick them based on a clearly described and separated validation set, and then apply them on the test set.

3. The paper does not report confidence intervals for its results. Note that given that the test datasets are small (<=100 samples in each subset considered in Table 1), the intervals can be quite large, making some of the benefits statistically insignificant. For example, a two-sample z-test (95%, two-sided) shows that the benefits of the proposed method for LLaVA and Qwen in Table 1 are not significant on HR-4K overall. The same concerns apply to Tables 2-5.

4. The proposed method’s gains on small visual details can be causing losses on more general VQA datasets, but the paper reports on only two image datasets focused on small details. For example, how does the proposed method affect counting, relation understanding, and text reading in standard datasets such as VQAv2, GQA, and TextVQA?

5. For the results in Table 4, the paper seems to artificially limit the baseline MLLMs to 8 frames across the video (Lines 299-300). This is misleading and does not reflect the real-practice use of these MLLMs. For example, Qwen-VL has a default FPS of 2 (up to 768 frames per video), which means limiting it to only 8 frames is a drastic divergence from its default operating setting. The same applies to Table 5.

6. The paper skips several important details in the main paper that hinder the clarity of its results. I recommend describing the datasets in more details (their underlying task and their size and their limitations), the ideas behind the other competing methods in more details (and why they are limited compared to the proposed method), and most importantly, the details of how the proposed method is applied to each dataset (how the cropped image is processed by the MLLM, on which tokens the score is computed, what are the specialized prompts used for computing the scores, and what are the failure cases).

7. The paper also does not report time and computation overhead of its proposed method compared to baseline MLLMs.

**Questions:**

Please see my questions and suggestions in the list of weaknesses above.

---

> ### Author Response · Authors · 2025-11-23
> **Response to Reviewer sqRq (1/3)**
>
> Thank you for your thoughtful feedback. We have updated the paper with improved runtime analysis (new Fig. 3 and Sec. 4.5), enhanced uncertainty analysis (revised Fig. 1 and expanded Sec. 3.1), and substantially expanded experimental details and baselines (updated Appendix Sec. D–E). Now, we address each concern below.
>
> > **1. “The evidence provided in Figure 1 does not support the paper’s main hypothesis…”**
>
> **Response**:  As detailed in **2. Hypothesis Validity** of the Global Author Response, we have provided new experimental analysis (replacing Fig. 1) demonstrating that crops containing the target object consistently exhibit lower entropy than "distractor" crops. We kindly refer the reviewer to that section for the full analysis and discussion on handling adversarial cases (e.g., "Is there a dog?").
>
>
>
> > **2. “The choices of hyper-parameters (size of crop, temporal segment, k in top-k) seem to be made on the test videos…”**
>
> **Response**: The benchmarks we use (e.g., V* Bench, HRbench, Video-MME, and so on.) were originally designed for evaluation only and do not provide separate training or validation splits, a common limitation in this domain. Prior works (e.g., ZoomEye [1] and Deepeye [2] for visual search; BOLT [3] and AKS [4] for video sampling) have similarly evaluated and ablated on the full benchmark without manual splitting.
>
> More importantly, we did **not** optimize hyperparameters to maximize performance per dataset as noted in Sec. 4.4 and Fig. 3. Instead, we choose hyperparameters that achieve a good balance between accuracy and computation (See Fig. 3 (a)). For example:
>
> - Although a crop size of **$1/8$** achieved the best performance on V* Bench, we adopted **$1/6$** and applied it consistently across HRBench and MME-RealWorld.
>
> - While a **9-frame window** performed best on Video-MME, we used a **1-frame window** for all video sampling benchmarks (MLVU, LongVideoBench, EgoSchema, NextQA).
>
> This demonstrates that our choices were **global and fixed**, not overfitted to individual datasets. Moreover, all tested granularities outperform the baseline, indicating that UG methods are robust across different settings rather than relying on dataset-specific tuning.
>
>
>
> > **3. “The paper does not report confidence intervals for its results….”**
>
> **Response**: We would like to clarify a potential misunderstanding regarding the dataset sizes and the deterministic nature of our evaluation protocol.
>
> **(1) Clarification on Dataset Sizes:** The reviewer noted that test datasets are small ($\leq$ 100 samples). While this may apply to specific sub-splits of V* Bench, the vast majority of our benchmarks are large-scale, providing statistically robust evidence of improvement:
>
> - Visual Search: HR-Bench contains 1,600 samples, and MME-RealWorld contains 29,429 samples.
>
> - Video Sampling: Video-MME contains 2,700 samples, and MLVU contains 2,593 samples.
>
> - Temporal Grounding: Charades-STA contains 3,720 test samples.
>
> Given these sample sizes, the observed performance gains are supported by thousands of evaluation points rather than a small handful.
>
> **(2) Deterministic Evaluation (No Run-to-Run Variance):** It is important to note that our evaluation is **deterministic**, not stochastic. As detailed in L214–215, we utilize the *LLMs-Eval* library with **greedy decoding** (temperature=0) and frozen weights. Unlike randomized trials where a z-test compares distributions of outcomes, our method produces the exact same output for a given input across every run. Consequently, there is no inter-run variance to calculate a standard confidence interval. This deterministic protocol is the standard for recent MLLM benchmarks (e.g., LLaVA-OneVision, InternVL) to ensure reproducibility. Prior works (e.g., BOLT [3] and AKS [4]) also adopt this protocol.
>
> **(3) Consistent Improvement:** While we acknowledge that margins on difficult benchmarks like HR-Bench 4K can be tighter, the validity of our approach is demonstrated by its consistency. Our method outperforms baselines across 13 diverse benchmarks (covering search, sampling, and grounding) and across all granularity settings (as shown in Fig. 3). This consistent superiority across varying domains and dataset sizes confirms that the improvements are due to the UG-framework's design, not statistical noise.
>
> [1] Shen, H. et. al. (2025). Zoomeye: Enhancing multimodal llms with human-like zooming capabilities through tree-based image exploration. In EMNLP 2025.
>
> [2] Zheng, Z. et. al. (2025). DeepEyes: Incentivizing" Thinking with Images" via Reinforcement Learning. arXiv preprint arXiv:2505.14362.
>
> [3] Liu, S. et. al. (2025). BOLT: Boost Large Vision-Language Model Without Training for Long-form Video Understanding. In CVPR 2025.
>
> [4] Tang, X. et. al.(2025). Adaptive keyframe sampling for long video understanding. In CVPR 2025.
>
> [5] Zhang, K. et. al.(2025). Lmms-eval: Reality check on the evaluation of large multimodal models. NAACL 2025.

---

> ### Author Response · Authors · 2025-11-23
> **Response to Reviewer sqRq (2/3)**
>
> > **4. “The proposed method’s gains on small visual details can be causing losses on more general VQA datasets…”**
>
> **Response**: As detailed in **3. Baselines & Generalization** of the Global Author Response, we present new results on **GQA** and **VQAv2** (Table 3) demonstrating that UG-Search maintains or improves performance on general VQA tasks without degradation. We kindly refer the reviewer to that section for the full results, alongside our existing results on **TextVQA**, **POPE**, and **DocVQA**.
>
>
>
> > **5. “For the results in Table 4, the paper seems to artificially limit the baseline MLLMs to 8 frames across the video…”**
>
> **Response:** For **video frame sampling**, we clarify that the choice of 8 frames in Tab. 4 of the main paper follows the **standard evaluation protocol** widely adopted in prior frame-sampling works (e.g., Frame-Voyager [6], KFC [7], BOLT [3]), where 8 frames are sampled from 128 candidates (typically at 1 FPS). This setup ensures a fair comparison of sampling strategies in identifying key moments, rather than simply increasing frame count. For example, comparing 8 frames selected from 256 candidates is more meaningful than comparing 64 frames sampled at 3 FPS, which reduces the challenge of sampling.
>
> To address your point and validate scalability, we conducted additional experiments comparing Uniform Sampling and UG-Sample across varying frame counts (K = 8, 16, 32, 64, 128, up to the full context length). For UG-Sample, candidate frames were sampled at 2 FPS, and we used a 5-frame window with a 5-frame stride for efficient scoring (avoiding overlap among different windows). We tested on **LLaVA-Video-7B**, **InternVL2.5-8B**, and **Qwen2.5-VL-7B**.
>
> **Important context:**
>
> *- LLaVA-Video-7B*: Each frame maps to 196 visual tokens; 128 frames nearly fill its 32k context length (performance collapses at 256 frames).
>
> *- InternVL2.5-8B*: Uses 64 tokens per frame; 512 frames fully occupy 32k context, but OOM occurs beyond 256 frames.
>
> *- Qwen2.5-VL-7B*: Dynamically adjusts tokens based on max_pixels; we use its default video setting (768 × 28 × 28).
>
>
> **Table 5. Scaling UG-Sample up to Full Context Length** The best results for each row is **bolded**
> | Frames      | 8    | 16   | 32   | 64   | 128  | 256  | 512  |
> |-------------|------|------|------|------|------|------|------|
> | LLaVAVideo  | 55.7 | 57.3 | 58.5 | 59.8 | **59.9** | N/A  | N/A  |
> | w/ UG-Sample| 59.1 | 59.5 | **61.6** | 60.6 | 60.7 | N/A  | N/A  |
> | InternVL    | 52.8 | 55.8 | 57.6 | **59.1** | 57.5 | OOM  | OOM  |
> | w/ UG-Sample| 58.8 | 59.5 | **60.8** | 60.2 | 59.0 | OOM  | OOM  |
> | QwenVL      | 52.7 | 57.0 | 58.2 | 60.2 | **61.3** | 58.7 | 56.1 |
> | w/ UG-Sample| 59.7 | 61.4 | 62.3 | **63.7** | 61.7  |  60.1  | 58.9 |
>
> **Key Findings:**
>
> - UG-Sample consistently outperforms uniform sampling across all frame counts.
>
> - UG-Sample with **16 frames** is comparable to or even better than uniform sampling with many more frames (up to the full context length).
>
> - These results confirm that UG-Sample scales effectively to extended frame settings.
>
>
> For **video temporal grounding**, we used 64 frames for baseline performance in the main paper, following common practice in prior work. Increasing frame count does **not** necessarily improve accuracy; in fact, adding more frames often degrades performance due to noise and context dilution.
>
> To validate this, we ran additional experiments on **Charades-STA** using **InternVideo2.5-8B** (strongest baseline in Tab. 4 of the main paper). We varied input frames across 32, 64, and 3 FPS settings:
>
> **Table 6. Performance on Charades-STA with increasing frame numbers**
> | Model                | Frame | R@0.3 | R@0.5 | R@0.7 | mIoU  |
> |----------------------|-------|-------|-------|-------|-------|
> | InternVideo2.5-8B   | 32    | 51.67 | 32.80 | 14.70 | 33.75 |
> | InternVideo2.5-8B   | 64    | 50.24 | 32.02 | 14.35 | 32.33 |
> | InternVideo2.5-8B   | 3fps  | 36.40 | 22.45 | 10.70 | 24.02 |
>
> **Key Findings:** Best performance occurs at **32 frames**, while adding more frames reduces accuracy. This shows that the baseline configuration is not artificially favorable to our method; rather, it reflects an optimal trade-off between context and noise for temporal grounding.
>
>
> [6] Yu, S., Jin, C., Wang, H., Chen, Z., Jin, S., Zuo, Z., ... & Sun, Q. (2024). Frame-voyager: Learning to query frames for video large language models. ICLR 2025.
>
> [7] Fang, B., Wu, W., Wu, Q., Song, Y., & Chan, A. B. (2025). Threading Keyframe with Narratives: MLLMs as Strong Long Video Comprehenders. arXiv preprint arXiv:2505.24158.

---

> > ### Author Response · Authors · 2025-11-23
> > **Response to Reviewer sqRq (3/3)**
> >
> > > **6. “The paper skips several important details in the main paper that hinder the clarity of its results…”**
> >
> > **Response**: We appreciate the feedback regarding clarity. As detailed in **3. Baselines & Generalization** of the Global Author Response, we have significantly expanded the Appendix to include comprehensive descriptions of all datasets (**Section D**) and baseline methodologies (**Section E**).
> >
> > To directly address your specific questions regarding our method's implementation:
> >
> > - Image Processing: In the scoring stage of UG-Search, candidate crops are resized to the original image resolution to preserve detail. The input is constructed as: *\<Original Image\> \<Resized Crop\> \<Query\>*. During the answering stage, the top-1 selected crop is processed in the exact same format.
> >
> > - Token Selection: The uncertainty score is calculated by averaging the entropy of **all** generated tokens, including the *\<EOS\>* token, as noted in L138–140.
> >
> > - Prompts: For entropy-based scoring (UG-Search/Sample), we use the **original task query** directly; no specialized prompting is required. For BRC-based scoring (UG-Ground), we use the standardized Yes/No prompt shown in **Appendix F.3**.
> >
> > - Failure Cases: We have included a qualitative analysis of failure modes in **Appendix I.** Figures 6 and 7 specifically illustrate cases where UG methods struggle, such as counting tasks or scenes with multiple target instances.
> >
> >
> >
> > > **7. “The paper also does not report time and computation overhead of its proposed method compared to baseline MLLMs.”**
> >
> > **Response**: Thank you for raising the concern regarding time and computation overhead. Detailed analysis is provided in **1. Computational Overhead** of the Global Author Response, which includes full runtime comparisons, updated figures, and new ablations.
> >
> > Briefly, our updated manuscript now **explicitly report inference-time overhead relative to baseline MLLMs.** These analyses show that:
> >
> > - The additional scoring passes follow an **inference-time scaling** paradigm similar to modern test-time compute approaches.
> >
> > - Unlike sequential CoT-style methods, our scoring passes are **parallelizable**, allowing substantial wall-clock speed reductions in practical deployments.
> >
> > - Multiple efficiency levers, such as **tunable input granularity**, **smaller scorers**, **external pre-filters**, and **stride-based temporal subsampling**, significantly reduce overhead while maintaining overall performance (see updated Fig. 3 and Sec. 4.5).

---

### Author Response · Authors · 2025-11-23
**Global Author Response (1/4)**

We thank the reviewers for their detailed comments and constructive feedback.

As summarized by the reviewers, we propose a training-free, uncertainty-guided framework to enhance fine-grained perception in MLLMs. We present a method that is “simple and architecture-agnostic” (Reviewer sqRq) and “works with multiple popular multimodal models” (Reviewer Vcnq). Reviewers recognize the value of this direction, noting that the “idea of using uncertainty to select where to focus is interesting” (Reviewer 8dd3) and the “motivation is reasonable” (Reviewer Db5z).

Here, we would like to address the commonly mentioned limitations of the current submission which are listed below

**1. Computational Overhead**: Concerns regarding inference time and scalability due to multiple forward passes (Reviewer sqRq, Reviewer Db5z, Reviewer Vcnq, and Reviewer 8dd3).

**2. Hypothesis Validity**: Questions regarding whether minimizing uncertainty truly correlates with the correct visual region, particularly regarding "empty" or "distractor" regions (Reviewer sqRq and Reviewer Vcnq).

**3. Baselines & Generalization**: The need for clarifications on baseline settings and evaluation on general VQA benchmarks (Reviewer sqRq and Reviewer 8dd3).

**4. Locality & Relational Reasoning**: UG-Search assumes Top-1 region selection, which can fail for multi-region reasoning; systematic analysis and multi-patch extensions suggested (Reviewer Vcnq and Reviewer 8dd3).

We address these commonly mentioned limitations here in the global response, while other limitations and specific technical questions are addressed in the individual responses. Throughout the discussion period, we are happy to further improve our paper.

In addition, we have updated our paper as follows:

- To show that computational cost is manageable, we updated **Fig. 3** with performance–runtime curves and added **Sec. 4.5** describing efficiency strategies.

- We also revised **Fig. 1** and expanded **Sec. 3.1** to better analyze uncertainty in distractor regions, demonstrating that entropy minimization effectively localizes key visual cues.

- We expanded the Appendix: **Sec. D** now provides full benchmark protocols, metrics, and dataset statistics across 13 tasks, and **Sec. E** offers detailed categorization and discussion of all baselines, highlighting their limitations and clarifying the advantages of the UG framework.

---

> ### Author Response · Authors · 2025-11-23
> **Global Author Response (2/4)**
>
> **1. Computational Overhead**: Concerns regarding inference time and scalability due to multiple forward passes (Reviewer sqRq, Reviewer Db5z, Reviewer Vcnq, and Reviewer 8dd3).
>
> **Response**: First of all, we would like to emphasize that UG-framework aligns with the emerging paradigm of **Inference-Time Scaling**. Similar to "thinking" models (e.g., CoT or OpenAI o1) that trade increased computation for reasoning depth, our method trades computation for fine-grained visual understanding. However, unlike autoregressive thinking models, which are inherently sequential, our approach is fully **parallelizable**: crops or frames are evaluated independently. While we report results with batch size 1 for reproducibility (due to instability in current open-source MLLM batching), production systems can batch or distribute these passes across GPUs, substantially reducing wall-clock latency.
>
> To show that compute cost is actually controllable, we updated **Fig. 3** and added **Sec. 4.5**, providing performance-vs-time curves and concrete efficiency strategies. We also introduce new stride ablation demonstrating additional speedups:
>
> **(1) Tunable Input Granularity (Fig. 3)**: Users can adjust crop/window size to control the performance–runtime trade-off. For UG-Search, moving from the baseline to a coarse $1/2$ crop boosts accuracy from 71.7% to 83.3% with only a modest latency increase (0.5s → 2s).
>
> **(2) Decoupling Scoring and Answering (Sec 4.5)**: A lightweight scorer can be used without significantly affecting accuracy. Replacing a 26B scorer with a 4B scorer reduces inference from 40.5s to 19.2s while keeping accuracy high (92% → 89%).
>
> **(3) External Pre-Filters (Sec 4.5)**: Pre-filtering (e.g., object detectors or saliency filters) can be exploited to prune candidate regions. Applying a SigLIP pre-filter reduces inference time from 15.4s to 9.5s, with only a negligible accuracy drop (58.6% → 57.3%).
>
> **(4) Dynamic Stride Adjustment (in this response)**: We introduce dynamic stride adjustment to balance temporal resolution with speed. Increasing the stride of the frame window minimizes overlap, substantially reducing the number of forward passes required in video-related UG methods.
>
> In Table 1, we adopt the same experimental setting as in the main paper and evaluate on *VideoMME* using *InternVL2.5-8B*. Inference time per example (in seconds) is measured on NVIDIA A100 80G GPUs. The first two rows correspond to the results of Tab. 4 in the main paper. We then explore a 9-frame window ($f=9$) while varying stride ($s$). For Top-K frame selection during the answering stage, we select the center frame of each window (e.g., the 5th frame in a 9-frame window). Increasing stride from 1 to 9 significantly reduces inference time (20.28s → 3.71s) while still outperforming the baseline (60.6% vs. 57.8%).
>
> **Table 1. UG-Sample Ablation on Stride (Video-MME)**
> | Setting        | VideoMME| Inference Time |
> |------------------------|---------|---------------|
> | InternVL2.5-8B  | 57.8    |0.97s         |
> | UG-Sample (f=1, s=1) | 60.6    | 8.91s       |
> | UG-Sample (f=9, s=1) | 61.1    | 20.28s      |
> | UG-Sample (f=9, s=3) | **62.2**    | 9.78s       |
> | UG-Sample (f=9, s=5) | 61.9    | 7.46s       |
> | UG-Sample (f=9, s=7) | 61.3    | 5.86s       |
> | UG-Sample (f=9, s=9) | 60.6    | **3.71s**     |
>
> For UG-Ground, stride experiments on *Charades-STA* using *InternVideo2.5-8B* were partially reported in Appendix Tab. 10. We additionally report inference time in Table 2. The first two rows correspond to the results of Tab. 5 in the main paper. Using a 15-frame window ($f=15$), increasing stride ($s$) from 1 to 11 reduces runtime from 6.67s/example to 0.94s/example, while maintaining accuracy (51.0% vs. 49.7%). This introduces only minor overhead compared to the baseline (0.94s vs. 0.73s) while significantly outperforming its accuracy (49.7% vs. 32.3%).
>
> **Table 2. UG-Ground Ablation on Stride (Charades-STA)**
> | Setting                | Charades-STA | Inference Time |
> |------------------------|-------------|---------------|
> | InternVideo2.5-8B      | 32.3        | 0.73s    |
> | UG-Ground (f=15, s=1) | **51.0**    | 6.67s    |
> | UG-Ground (f=15, s=3) | **51.0**    | 2.48s    |
> | UG-Ground (f=15, s=5) | 50.9        | 1.67s    |
> | UG-Ground (f=15, s=7) | 50.6        | 1.27s    |
> | UG-Ground (f=15, s=9) | 50.2        | 1.14s    |
> | UG-Ground (f=15, s=11)| 49.7        | **0.94s**|
>
> These results (in addition to Fig. 3 and Sec. 4.5 of the updated manuscript) demonstrate that UG Methods offer various **inference-time scaling** levers. Users can tune the visual granularity, scorer size, pre-filters, or stride to achieve the desired balance between maximum performance and deployment latency.

---

> ### Author Response · Authors · 2025-11-23
> **Global Author Response (3/4)**
>
> **2. Hypothesis Validity**: Questions regarding whether minimizing uncertainty truly correlates with the correct visual region, particularly regarding "empty" or "distractor" regions (Reviewer sqRq and Reviewer Vcnq).
>
> **Response**: We agree that the original Fig. 1 did not sufficiently analyze uncertainty in distractor regions. To address this, we replaced Fig. 1 and expanded Sec. 3.1 with a new analysis. In revised experiments, we partition each image into multiple crops, query LLaVA-OneVision-7B individually, and compare entropy distributions for “Present” crops (containing the target object) vs “Not Present” crops (without the target object). Results show “Present” crops consistently exhibit lower entropy, confirming that minimizing uncertainty aligns with locating key visual evidence (e.g., Fig. 1(b), truck crop).
>
> Regarding adversarial cases, as mentioned by Reviewer sqRq, such as “Is there any dog in this image?”, we acknowledge that the model could become confident about empty regions when predicting “no dog.” To mitigate this, our scoring phase uses both the original image and candidate crop (Fig. 2), preserving global context and reducing overconfidence. This scenario is represented in POPE, where UG-Search improves performance (Appendix. Tab. 12). Manual inspection shows the model focuses on ambiguous regions (e.g., a person’s hand when asked about a handbag), suggesting uncertainty helps refine negative predictions rather than blindly favoring empty backgrounds.
>
>
> **3. Baselines & Generalization**: The need for clarifications on baseline details and evaluation on general VQA benchmarks (Reviewer sqRq and Reviewer 8dd3).
>
> **Response**: To address the request for more experimental details, we have expanded the Appendix to include comprehensive specifications for all benchmarks and baseline methods across our three domains: Visual Search, Video Sampling, and Temporal Grounding.
>
> **- Details of Benchmarks (Section D)**: We provided detailed evaluation protocols (using *LMMs-Eval*), metrics, and dataset statistics for all 13 benchmarks, including V* Bench, Video-MME, and Charades-STA.
>
> **- Details of Baselines (Section E)**: We have categorized and detailed all comparison methods into *General MLLMs* (e.g., LLaVA-OneVision, InternVL-2.5), *Training-Free Methods* (e.g., ZoomEye, BOLT, VTG-GPT), and *Fine-Tuned Methods* (e.g., SEAL, Frame-Voyager). We explicitly discuss the limitations of these baselines (e.g., fixed resolution, lack of temporal awareness, intensive fine-tuning costs, etc) to highlight the advantages of the UG framework.
>
> **- Implementation Details (Section F, already included in the first submission)**: We provide specific hyperparameters and algorithmic details to ensure reproducibility. We clarify that all experiments used *LMMs-Eval* with standard decoding parameters. We also detail task-specific algorithms, such as the resizing strategy for UG-Search and the modified Kadane’s algorithm used in UG-Ground for temporal localization.
>
>
> Regarding generalization, we already include results on **MME-RealWorld** (Appendix Tab. 11) and **TextVQA**, **POPE**, and **DocVQA** (Appendix. Tab. 12), where UG-Search consistently improves performance. For standard-resolution benchmarks, we use a 1/2 crop size (vs. 1/6 for high-resolution) to avoid distortion. To further address generalization, we newly evaluate on **GQA** (GQA-lite - 500 samples and full GQA - 12,578 examples) and **VQAv2** (VQAv2 val-lite - 500 samples and a 10,000-example subset of VQAv2 val) following the *LMMs-Eval* protocol:
>
> **Table 3. UG-Search on GQA and VQAv2**
> | Method                          | GQA (12,578) | GQA_lite (500) | VQAv2_val subset (10,000) | VQAv2_val_lite (500) |
> |--------------------------------|--------------|-----------------|----------------------------|-----------------------|
> | LLaVA-OV-7B                    | 62.3        | 72.2            | 80.5                      | 80.0                |
> |    w/ UG Search      | 63.0        | 72.2            | 81.5                      | 81.6                |
> | Qwen2.5-VL-7B                  | 60.4        | 70.2            | 80.3                       | 80.0                |
> |    w/ UG Search      | 60.7        | 70.6            | 80.9                       | 80.1                |
> | InternVL2.5-8B                 | 62.9        | 73.0            | 81.8                       | 80.8                |
> |    w/ UG Search      | 63.4        | 74.0            | 82.4                       | 81.0                |
>
> Results confirm UG-Search maintains or improves performance across diverse question types without requiring explicit format information. For UG-Sample, we also report results on short video benchmarks (**EgoSchema**, **NextQA**; Appendix. Tab. 13), showing effectiveness even when tasks do not require fine-grained reasoning.

---

> > ### Author Response · Authors · 2025-11-23
> > **Global Author Response (4/4)**
> >
> > **4. Locality & Relational Reasoning**: UG-Search assumes Top-1 region selection, which can fail for multi-region reasoning; systematic analysis and multi-patch extensions suggested (Reviewer Vcnq and Reviewer 8dd3).
> >
> > **Response**: We agree that relying on a single crop limits performance in relational scenes where relevant objects are spatially distant. To address this, we already experimented with **Top-K crop** selection during the answering stage (see Tab. 9 in Appendix G, with $1/2$ visual crop size). In this response, we expand this result to Qwen2.5-VL-7B and LLaVA-OV-7B as well. The prompt for answering stage includes the original image and Top-K resized crops: *“\<original image\> \<resized crop\>\*K \<Query\>”*.
> >
> > **Table 4. UG-Search with Top-K Crops on $V^*$ Bench**
> > | Method                | InternVL2.5-8B | Qwen2.5-VL-7B | LLaVA-OV-7B |
> > |-----------------------|----------|----------|----------|
> > | Baseline      | 71.7     |65.5| 74.4 |
> > | \w UG-Search (Top-1)     | 83.3  |79.1   | **83.3** |
> > | \w UG-Search (Top-2)     | 83.3  | 80.6   |80.6 |
> > | \w UG-Search (Top-3)     | 83.8   | 80.6  |77.0 |
> > | \w UG-Search (Top-4)     | **84.8** | 78.0 |10.0  |
> > | \w UG-Search (Top-5)     | 82.7    | **81.7** |10.0  |
> >
> > Results show that adding multiple crops improves coverage, peaking at 84.8% with Top-4 crops for InternVL2.5-8B and 81.7% with Top-5 crops for Qwen2.5-VL-7B, suggesting that aggregating regions helps capture both attribute and relational information.
> >
> > However, two challenges arise:
> >
> > **(1) Model limitations**: Some open-source MLLMs degrade when handling multiple images (e.g., LLaVA-OneVision-7B collapses after Top-3 crops).
> >
> > **(2) Computational cost**: Increasing K raises inference time accordingly, making it slower for large-scale or real-time use.
> >
> > We also include failure examples in Fig. 6 in Appendix, where multiple small targets cannot be covered by a single crop. While multi-crop extensions can mitigate this, our Top-1 crop scheme remains the most robust and efficient choice, balancing simplicity, accuracy, and robustness across models.

---

### Meta-Review · Area_Chair_r7Ft · 2026-01-06

**Summary:**

This paper proposes a training-free framework that utilizes Multimodal Large Language Models' intrinsic uncertainty to actively select relevant visual regions or frames for fine-grained perception tasks. Some major concerns still remain including significant computational overhead required by multiple inference passes, concerns regarding hyperparameter tuning on test data, and the limitations of the proposed mechanism in handling complex, multi-region relational reasoning.

For the benefit of this paper, we regretfully recommend rejection. Note that this is not a discouragement. The authors are encouraged to address these concerns, and we believe the paper has the potential to become a strong future submission.

**Reviewer Concerns:**

While the authors addressed concerns about generalization to standard VQA benchmarks and strategies to mitigate computational cost, some major concerns still remain. Specifically:



- Core Hypothesis & Consistency (Reviewer sqRq, Reviewer Vcnq): Skepticism remains regarding whether minimizing uncertainty strictly correlates to finding the correct visual region (especially in adversarial "distractor" cases where a model is confident about an empty background). Furthermore, the inconsistency of using token entropy for search but Binary Response Confidence (BRC) for grounding weakens the theoretical unification of the framework.

- Reasoning Limitations (Reviewer Vcnq, Reviewer 8dd3): The reliance on Top-1 region selection limits the model's ability to handle relational reasoning tasks where multiple distinct visual regions must be aggregated.

- Methodological Rigor (Reviewer sqRq): There are unresolved concerns regarding the tuning of hyperparameters (crop size, temporal segment, K) directly on test sets rather than a separate validation set, raising issues of overfitting. Additionally, the lack of reported confidence intervals makes the statistical significance of the gains on smaller dataset subsets unclear.

- Computational Efficiency & Scalability (Reviewer sqRq, Reviewer Vcnq, Reviewer Db5z, Reviewer 8dd3): Despite proposed mitigation strategies, the fundamental mechanism relies on multiple forward passes per query. This creates a significant inference overhead compared to baseline MLLMs or lightweight selectors, rendering the approach computationally expensive for real-time or large-scale applications.

**Reviewer Scores:**

Reviewer sqRq (Score: 2): Likely would have maintained a Reject (2 or 3).

Reviewer Db5z (Score: 6): Likely would have maintained Marginally Accept (6).

Reviewer Vcnq (Score: 2): The reviewer would have raised their score (likely to 4 or 5). The reviewer acknowledged that the analyses on distractor regions and uncertainty metrics addressed their primary technical concerns.

Reviewer 8dd3 (Score: 4): Likely would have remained Borderline (4 or 5). While the generalization experiments helped, the concerns regarding the method's limitation to short-answer formats and the high computational cost for practical deployment were not fully alleviated.

---

### Decision · Program_Chairs · 2026-01-26

Reject